# Exploring the contribution of risk factors on major illness: a microsimulation study in England, 2023-2043

Anna Head [1] ✉, Ann Raymond[2], Laurie Rachet-Jacquet[2], Adam Briggs[2], Brendan Collins[1], Max Birkett[1], Anita Charlesworth[2], Martin O'Flaherty[1], Toby Watt [2,3] & Chris Kypridemos [1,3]

Multimorbidity is projected to continue increasing in England and many other countries. Here, we use a validated microsimulation model to quantify the potential impact of improving exposure levels of eight risk factors on the burden of major illness among adults aged 30+ in England between 2023-2043. We find that the biggest contributors to incident major illness are body mass index, smoking, systolic blood pressure, and physical inactivity. Theoretical minimum risk exposure levels of all risk factors could reduce 2043 major illness prevalence by 2 percentage points (95% uncertainty intervals: 1.3, 2.7) compared to the continuing trends (base-case) scenario; under a 10% improvement in all risk factors, we project a 0.3 percentage points (0.2, 0.4) reduction in major illness. The impact on health inequalities is mixed. Our findings show that large improvements in risk factors are unlikely to substantially reduce the major illness burden by 2043 due to population ageing.

The absolute numbers and proportion of adults in England living with multimorbidity or complex chronic conditions have been rising over the past decades and are likely to continue increasing[1,2], with similar patterns seen worldwide[3,4]. This has profound implications for health and social care systems in terms of the likely demand for services and resources, as well as the wider societal costs, such as reduced productivity. Given that many chronic conditions are lifelong, preventing or postponing their onset is essential for reducing future levels of multimorbidity.

There is substantial evidence of associations between sociodemographic risk factors, such as age and deprivation, and the incidence and prevalence of multimorbidity[5,6]. Moreover, many long-term conditions like cancers and cardiovascular disease (CVD) share common risk factors, such as smoking, physical inactivity, and unhealthy diets[7]. These behavioural risk factors are distributed unequally across sociodemographic groups[8] and may also be associated with the onset of multimorbidity[9], the speed of accumulation of multiple conditions[10], and subsequent health outcomes[11]. It is reasonable, then, to assume that reducing population-level risk

factor exposures will not only decrease the incidence of individual chronic conditions but also reduce multimorbidity and health inequalities.

Existing research estimates the direct effect of multiple risk factor reductions on the burden of specific conditions, such as Alzheimer's disease[12], or the effect of a specific risk factor on a group of causally-related disease outcomes, such as the effect of obesity on diabetes and CVD[13]. Five previously published simulation models simulate the effects of changing trends in risk factors on mortality and life expectancy: three globally, one Europe-wide, and one in France[14–18]. To our knowledge, the impact of changing the prevalence of multiple risk factors on multimorbidity or health inequalities has not yet been explored. Moreover, previous projections of changing trends in risk factors often assume no effect on case fatality; therefore, any reduction in the disease incidence rate results in a corresponding reduction in the prevalence. There is, however, evidence that a change in risk factor exposure also affects the survival of those living with chronic conditions[19,20], counterbalancing or potentially reversing the expected reduction in prevalence.

[1]Department of Public Health, Policy and Systems, University of Liverpool, Liverpool, UK. [2]The Health Foundation, London, UK. [3]These authors contributed equally: Toby Watt, Chris Kypridemos. ✉e-mail: anna.head2@liverpool.ac.uk

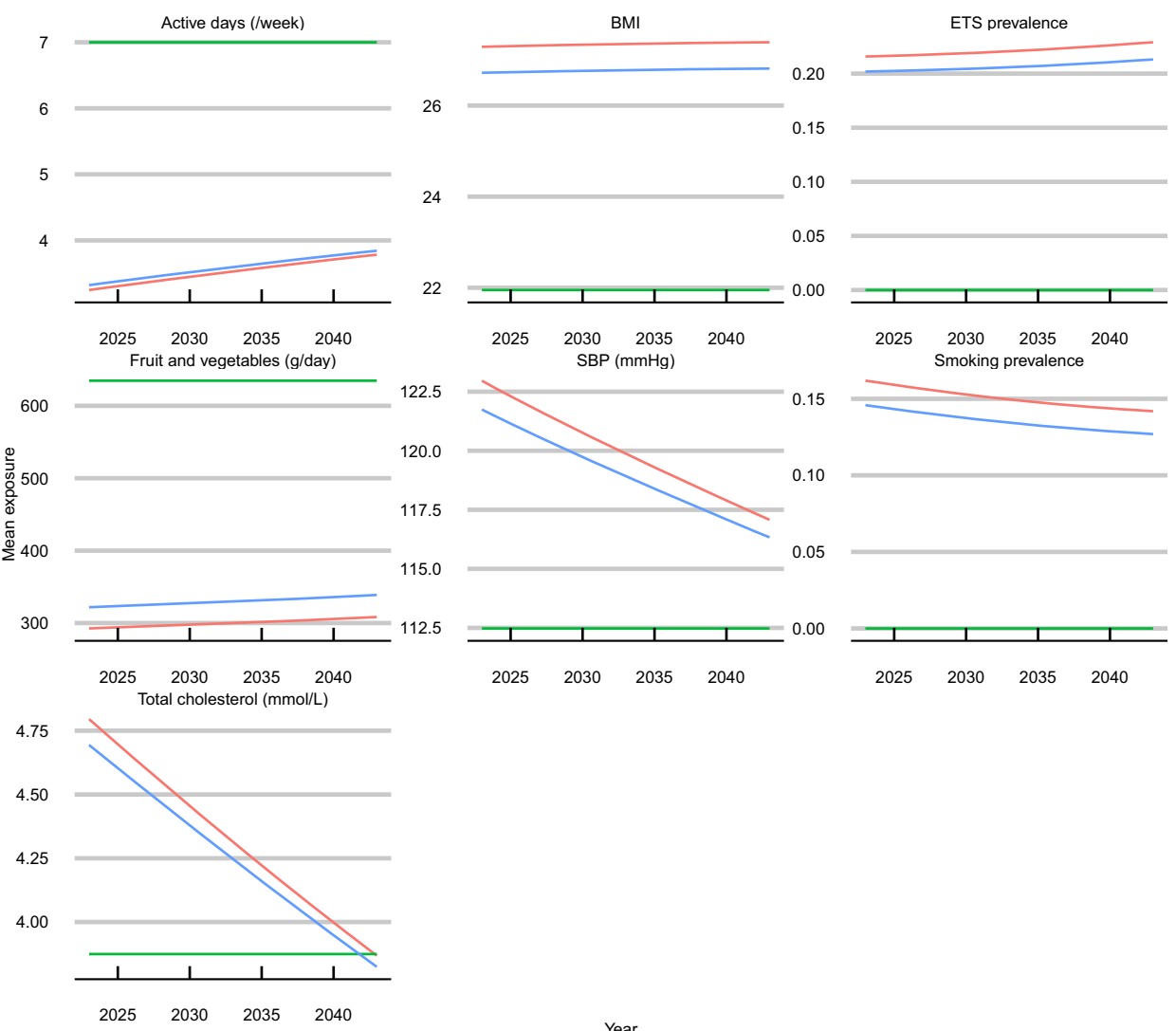

**Fig. 1 | Mean exposure levels (unstandardised) under each of the scenarios.**

Scenario — Continuing trends — Theoretical Minimum Exposure — 10% improvement

Fruit and vegetable consumption has been combined into a single scenario; ETS (environmental tobacco smoke) exposure is estimated based on smoking prevalence. BMI body mass index, SBP systolic blood pressure. Source data are provided as a Source data file.

The nature of public health prevention interventions makes them difficult to evaluate[21,22]. Simulation models are a cost-effective way to simulate the overall impact on health outcomes of risk factor scenarios. An understanding of which risk factors have the greatest potential for reducing disease burden, mortality, and health inequalities can be used to support public health policy design and implementation.

In this work, we estimate the effect of improving population risk factor exposures on future trends in major illness and mortality among the overall adult population of England and for population sub-groups living in different areas of deprivation. We use IMPACT$_{NCD}$, a validated dynamic discrete-time microsimulation model[23–27], to simulate the potential impact of two risk factor improvement scenarios on the burden of major illness in England among adults aged 30+ over the next two decades (2023–2043): (1) if the whole population had risk factor exposure levels at the level of estimated theoretical minimum risk; (2) if risk factor exposure levels improved by 10% relative to our base-case scenario. Our base-case scenario continues recent trends in risk factor exposures, as derived from 2003 to 2014 Health Survey for England (HSE) data. To explore the equity impacts of the scenarios, results for the theoretical minimum risk scenario are stratified by quintile groups of socioeconomic status using the English Index of Multiple Deprivation (IMD), an area-level measure of relative deprivation. We focus on eight risk factors (combined into six scenarios): tobacco smoking, environmental tobacco smoke, fruit consumption, vegetable consumption, physical activity, body mass index (BMI), systolic blood pressure (SBP), and total cholesterol. We define major illness as a score greater than 1.5 in the Cambridge Multimorbidity Score (CMS)—a composite index of 20 long-term conditions weighted based on their impact on the health system and an individual's health outcomes[28]. IMPACT$_{NCD}$ is informed by data from linked primary care, secondary care, and mortality records, risk factor exposure data from the HSE, and population estimates and projections from the Office for National Statistics (ONS). Further details on the modelling approach can be found in the 'Methods' and Supplementary Methods sections.

## Results

### Trends in risk factor exposure levels

Figure 1 displays the mean projected risk factor trends in England under each of the modelled scenarios—base-case of continuing trends, theoretical minimum risk exposure levels, 10% improvement—from

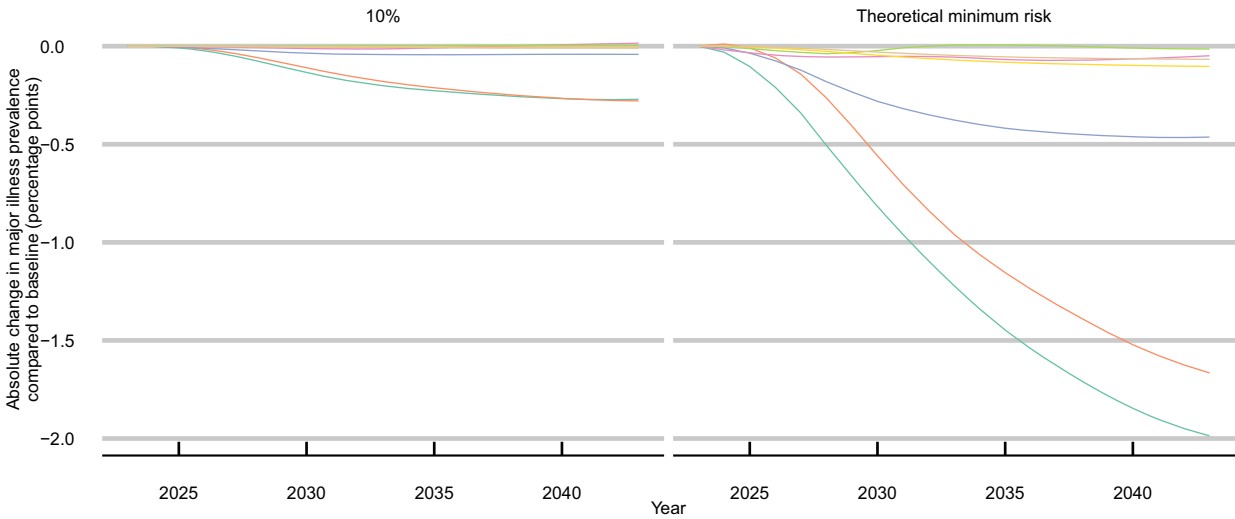

**Fig. 2 | Absolute difference (percentage points) in prevalence compared to the base-case scenario of continuing trends by risk factor and level of improvement.**

All curves are smoothed; for all scenarios (including base-case), the prevalence of major illness in 2023 was 25.7% (25.2, 26.1); Base-case scenario prevalence in 2043 was 29.8% (29.2, 30.4). SBP systolic blood pressure, BMI body mass index. Source data are provided as a Source data file.

Legend: All — BMI — SBP — Smoking — Physical activity — Fruit & veg — Total cholesterol

2023 to 2043. Risk factor trends by quintile groups of the IMD (an area-level measure of relative deprivation) and select age groups are presented in Supplementary Results Figs. 1.1 and 1.2.

**Theoretical minimum risk factor exposure levels**

If all risk factor levels within the population were at the theoretical minimum risk level from 2023, this would reduce the prevalence of major illness among adults aged 30 and over in 2043 by 2 percentage points (pp; 95% uncertainty interval: 1.3 pp, 2.7 pp), to 27.8% (27.3%, 28.4%), compared to 29.8% (29.2%, 30.4%) under a base-case scenario of continuing trends (Fig. 2); a relative reduction of 6.6% (4.5%, 9%) (Supplementary Results Table 3.1).

Setting each risk factor in turn to the theoretical minimum risk level of exposure suggests that the biggest contribution to the onset of major illness is from BMI, followed by smoking, SBP, and physical inactivity (Fig. 3A). The size of the reductions in major illness onset is driven by a combination of the prevalence of the risk factor exposure in the population (Fig. 1) and the number and strength of causal relationships modelled between each risk factor and the incidence of individual conditions (Table 2). Similarly, the differing lag times between exposure and incidence (Supplementary Methods p12) mean the impact on incidence rates from improved BMI is not likely to be seen until later than for other risk factors such as SBP. Tables of absolute and relative differences are provided in Supplementary Results Sections 4 and 7.

The decrease in projected all-cause mortality rate (Fig. 3B) under a scenario where all risk factors are at the theoretical minimum risk level is greatest 10 years after scenario implementation and then lessens to a reduction of 16.3 deaths per 10,000 persons at risk in 2043 (12.1, 20.2). This is partly driven by the base-case scenario improving trends for most risk factor exposures (Fig. 1). These reductions in incidence and mortality rates would prevent or postpone 5.5 m case-years of major illness between 2023 and 2043 (2.5 m, 8.6 m) (Fig. 3C) and increase the number of individuals living without major illness in 2043 by 2.1 million (1.8 m, 2.5 m) compared to the base-case scenario of continuing trends (Fig. 3D). For the smoking and physical activity scenarios, reductions in mortality rates are greater than in major illness incidence rates in some years, leading to a greater number of case-years lived with major illness when compared to the continuing trends scenario (Fig. 3C) as well as greater numbers of individuals living without major illness (Fig. 3D).

Figure 4 shows the change in prevalence of major illness relative to those in 2023 with and without changes to population ageing, illustrating the impact of demographic changes on the results. Presenting results scaled to the 2023 population structure ('excluding population ageing'), however, removes the impact of these scenarios on improving life expectancy. The projected population structures for selected years under the base-case scenario are presented in Supplementary Results Section 2.

**10% improvement in risk factor exposure levels**

Following a 10% improvement in all modelled risk factor exposure levels over 20 years, the projected prevalence of major illness could be 0.3 percentage points (0.2, 0.4) lower than under the base-case continuing trends scenario (Fig. 2; Supplementary Results Table 3.1). The effect on mortality rates differs by risk factor (Fig. 5B): under the 10% improvement in the SBP scenario, reductions in mortality rate are greatest 6 years after scenario implementation. Under the BMI and smoking 10% improvement scenarios, the effect on the reduced mortality rate is greatest after 10 years. The combined effect of decreased major illness incidence (Fig. 5A) and all-cause mortality rates (Fig. 5B) under the 10% improvement in all risk factors scenario would save 0.9 m (0.4 m, 1.4 m) years lived with major illness between 2023 and 2043 (Fig. 5C), and result in a steady increase in the numbers living without major illness (Fig. 5D).

**Impact on inequalities**

When stratified by quintile groups of the IMD (an area-level measure of relative deprivation), the impact on the 2043 projected prevalence of major illness has differential effects depending on the risk factor (Fig. 6). For this analysis, we only used the theoretical minimum risk factor exposure scenario. The 10% improvement scenario is relative to the base-case exposure level, and therefore, it implicitly assumes a differential effect by IMD group that renders its distributional effect less useful.

For BMI, greater absolute reductions are seen among people living in the most deprived compared to the least deprived quintile group. For example, a scenario of no excess BMI would reduce major illness prevalence amongst individuals in the most deprived IMD quintile group by 0.4 pp (0.1 pp, 0.6 pp) more compared to the least deprived quintile group. This socioeconomic gradient was observed in 99% of our simulations. Conversely, theoretical minimum risk levels of

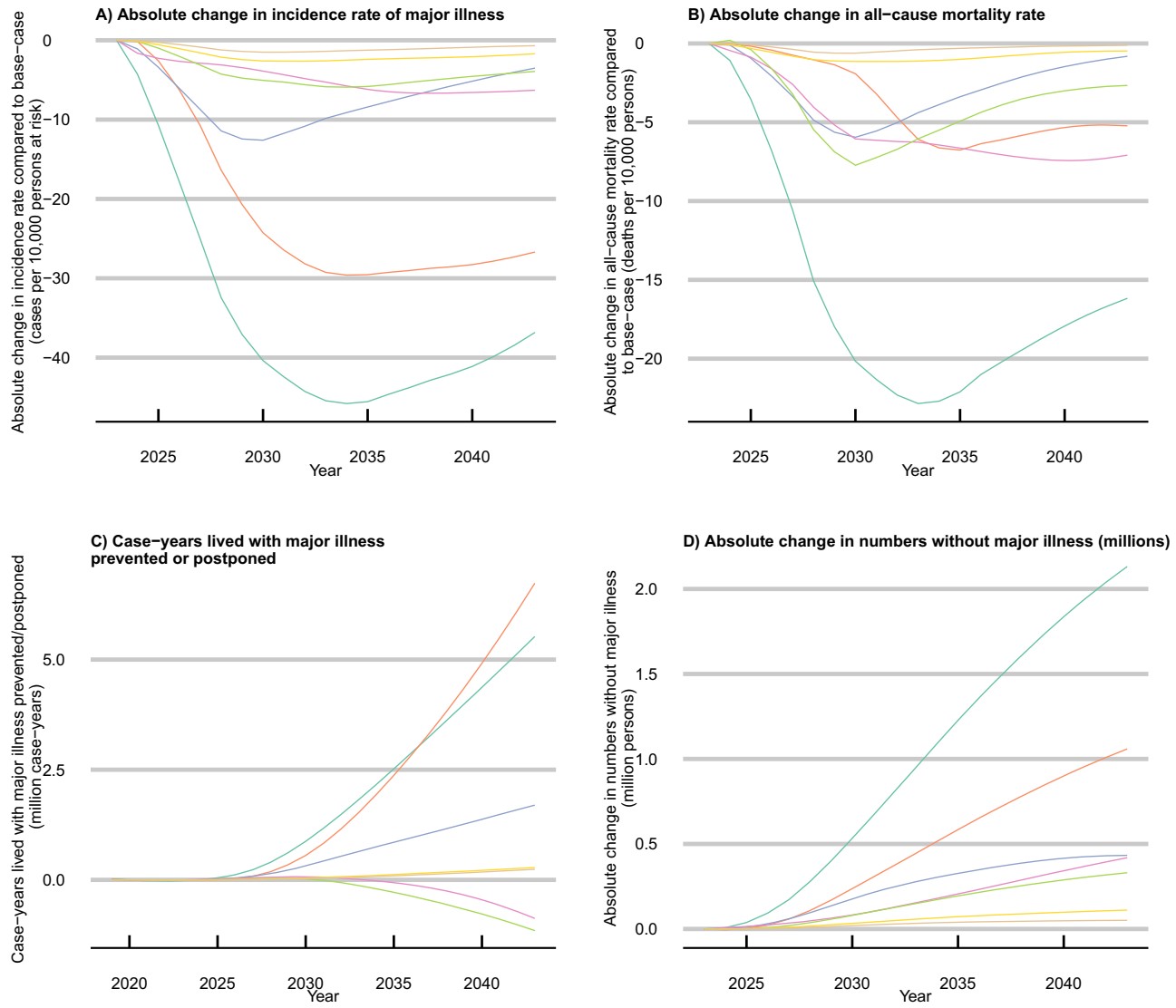

**Fig. 3 | Theoretical minimum risk factor exposure levels.** Absolute change in **A** projected incidence rates of major illness, **B** all-cause mortality rates, **C** case-years lived with major illness prevented or postponed, and **D** adults living without major illness under a scenario of theoretical minimum risk factor exposure levels. Case-years prevented/postponed are calculated from 2023 to 2043; All curves are smoothed. *Major illness is defined as a Cambridge Multimorbidity Score > 1.5. SBP systolic blood pressure, BMI body mass index. Source data are provided as a Source data file.

Legend: All — BMI — SBP — Smoking — Physical activity — Fruit & veg — Total cholesterol

SBP result in greater decreases in major illness prevalence among people living in the least deprived quintile group (by 0.4 pp (0.1 pp, 0.6 pp) more compared to the most deprived IMD quintile group. The socioeconomic gradients were small for physical activity, smoking, and total cholesterol; however, for all four of these factors, the reduction in major illness prevalence was more likely to be greater in the least deprived quintile group. We did not observe a gradient for the fruit and vegetable scenario. In particular for smoking, its elimination could lead to an increase in major illness prevalence among those living in the most deprived IMD quintile group, resulting from greater decreases in all-cause mortality than in major illness incidence—people living longer but at risk of more years in ill health.

### Impact by cohort

Twenty years is a relatively short timeframe to see the impact of improved risk factor levels at a population level, particularly as we modelled 14 out of the 20 conditions as lifelong. To explore the potential longer-term impacts, we present the absolute change in

prevalence for three cohorts: those born in 1989–1993 (aged 30–34 in 2023), 1969–1973 (aged 50–54 in 2023) and 1949–1953 (aged 70–74 in 2023). For most risk factors, the greatest absolute reductions in prevalence are likely to occur for the middle cohort (70–74 in 2043); for the youngest cohort, prevalence reductions do not occur until the end of the projection period (Supplementary Results Fig. 6.1). For the oldest cohort, the absolute reduction in the projected prevalence decreases in the final 5–10 years of the projection period because the relative risks of these exposures decrease in older age.

### Impact on population size and people living with major illness

Under our base-case scenario of continuing trends, the overall population size (aged 30 and over) is projected to grow by 6.3 million between 2023 and 2043[29], with an additional 3.4 million people projected to live with major illness in 2043 compared to 2023. Both the 10% improvement and theoretical minimum risk scenarios would further increase the total projected population size in 2043 compared to the base-case scenario, as well as reduce the increase in people living

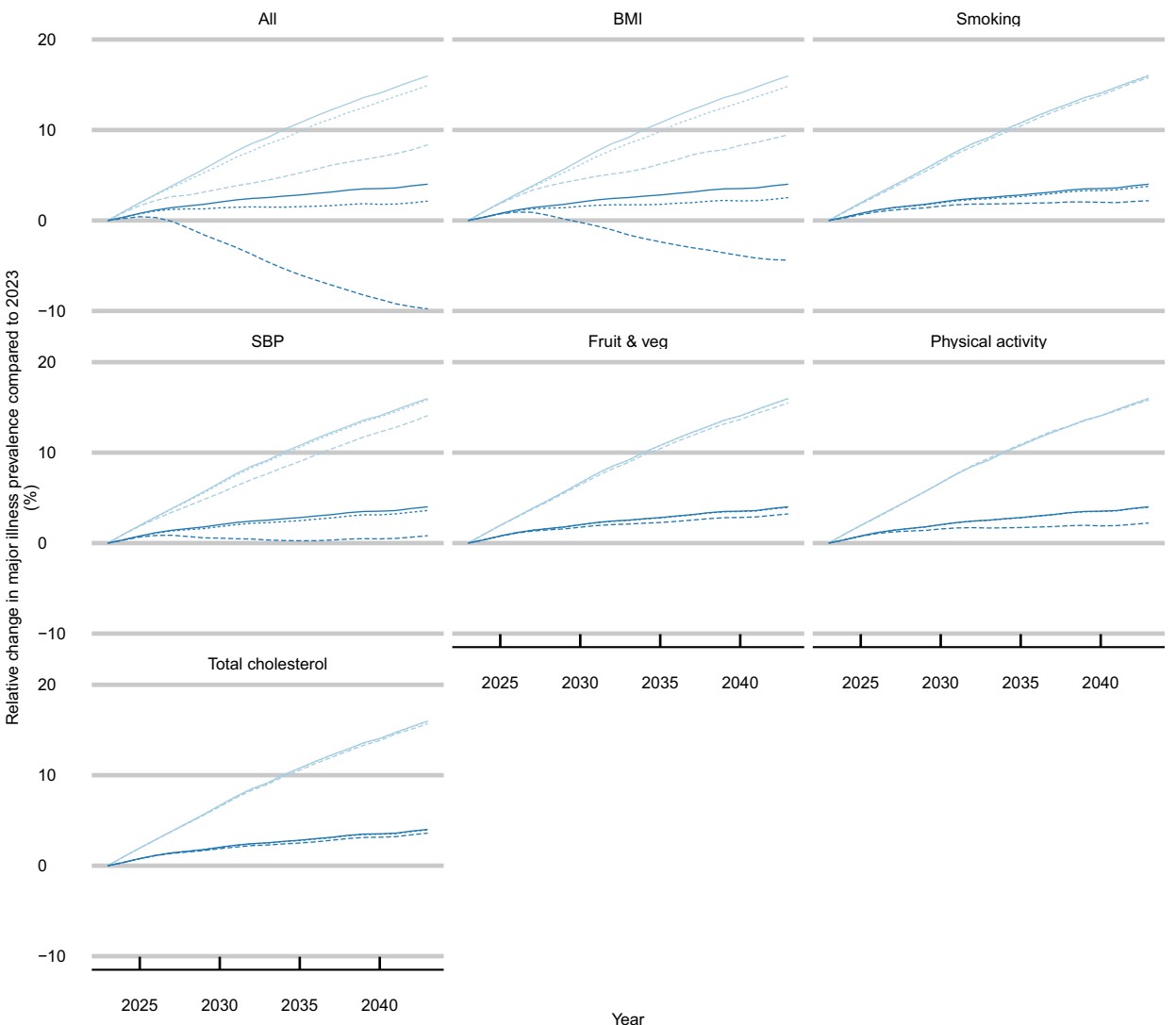

**Fig. 4 | Projected change in prevalence of major illness by scenario relative to estimated 2023 prevalence (25.7%), including and excluding population changes\*.**

\*Change rates excluding population changes were calculated by applying projected prevalence under each scenario to the 2023 population age structure; Major illness is defined as a Cambridge Multimorbidity Score >1.5; SBP systolic blood pressure, BMI body mass index. Source data are provided as a Source data file.

Including demographic changes — Excluding demographic changes — Scenario — Continuing trends ⋯⋯ 10% ----- Theoretical minimum risk

with major illness (Fig. 7). Our improvement scenarios represent a combination of primary and secondary prevention, and we have assumed that improvements in risk factor exposures affect case fatality and disease incidence. This results in a greater increase in numbers without major illness than the decrease in numbers with major illness under our improvement scenarios.

### Sensitivity analyses

Similar patterns in the projected prevalence of major illness were found in our main results compared to three sensitivity analyses: (1) assuming improved risk factor exposures did not impact case fatality, (2) adjusting our threshold CMS score for major illness, (3) modelling the direct effect only of a 10% improvement in BMI, excluding the indirect effect through improved SBP and total cholesterol (see Supplementary Results Section 7).

### Discussion

We used a validated microsimulation model to quantify the potential impact of improving exposure levels of eight risk factors on the burden of major illness among adults aged 30 and over in England between

2023 and 2043. Our results suggest that a 10% improvement in all modelled risk factors combined could potentially reduce the prevalence of major illness among adults aged 30 and over by 0.3 percentage points (95% UI: 0.2,0.4) compared to projected levels of ill health should current patterns in risk factor exposure continue: a 0.9% relative reduction (0.5%, 1.3%), and 220,000 fewer people living with major illness (180,000, 270,000). Despite this meaningful reduction, this is not sufficient to reverse the growing burden of major illness driven by an ageing population: the absolute prevalence of major illness is likely to remain high. Of the risk factors we modelled, BMI, SBP and smoking make the biggest contributions to the prevalence and incidence of major illness in England. Reducing BMI has the biggest impact on reducing major illness prevalence in individuals living in the most deprived IMD quintile group, whilst reducing SBP has the greatest impact among those in the least deprived quintile group.

The IMPACT$_{NCD}$ model is an advanced, validated, flexible microsimulation of the dynamics of risk factor exposures and non-communicable diseases (NCDs) in the adult population of England. The modelling of individual conditions allows in-depth exploration of how trends in eight risk factors may impact the future burden of major

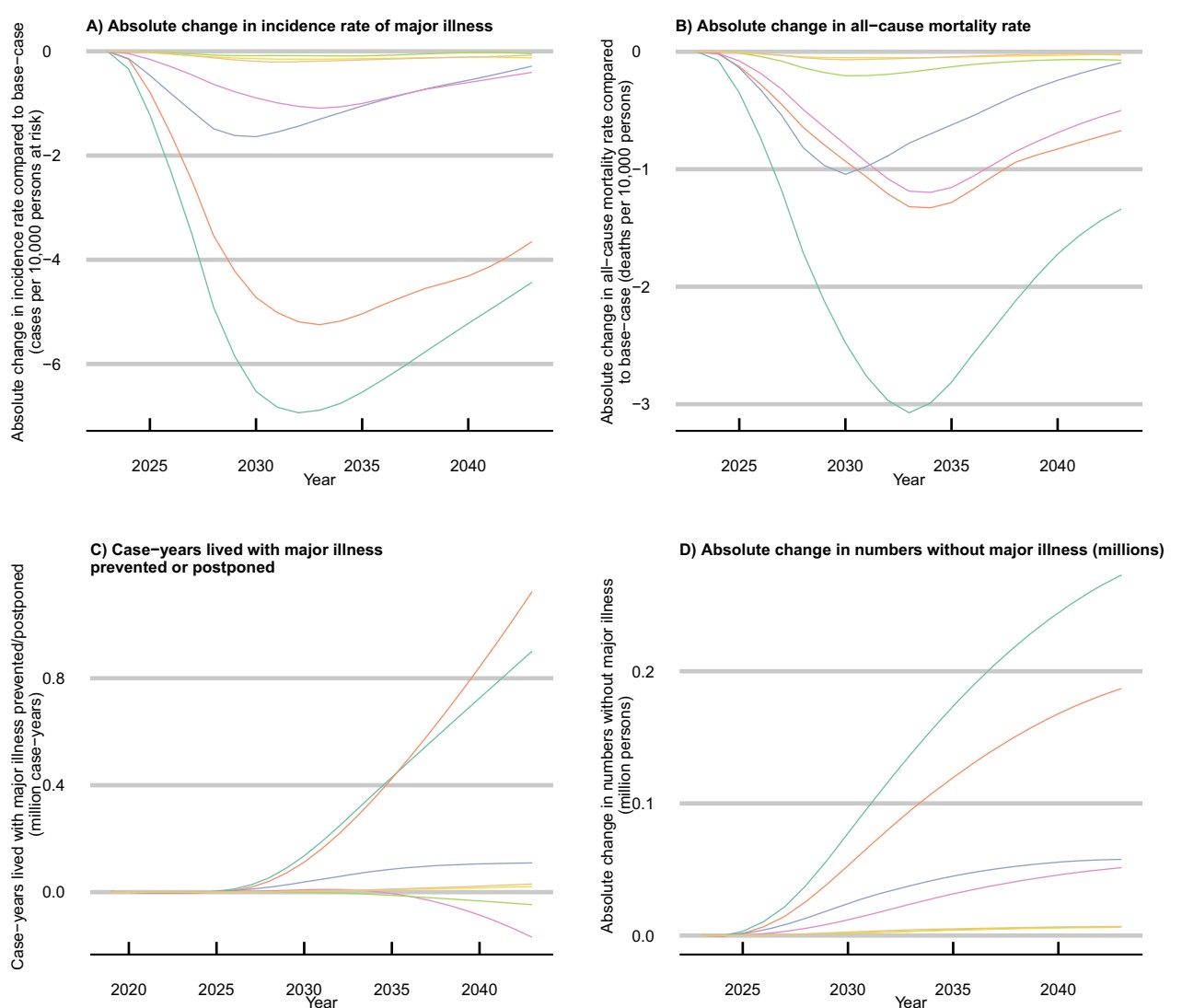

**Fig. 5 | 10% improvement in risk factor exposure levels.** Absolute change in (**A**) projected incidence rates of major illness 5, **B** all-cause mortality rates, **C** case-years lived with major illness prevented or postponed, and **D** adults living without major illness under a scenario of 10% improvement in risk factor exposures.

Case-years prevented/postponed are calculated from 2023 to 2043; all curves are smoothed. *Major illness is defined as a Cambridge Multimorbidity Score >1.5; SBP systolic blood pressure, BMI body mass index. Source data are provided as a Source data file.

illness. Our approach (see 'Methods' and Supplementary Methods for details) uses population-attributable risk fractions to translate risk factor exposures to disease incidence and case fatality and includes lag times between exposure and outcomes. We modelled dependencies between certain diseases based on the strength of the correlations and clinical understanding. The complex dynamic generated by improvements in risk factors and disease trends includes competing causes of illness and death. It facilitates realistic estimates of the future burden of illness from a societal perspective.

We modelled improved non-cause-specific mortality for improvements in physical activity, smoking, and SBP. In our sensitivity analysis, modelling the effect of improved risk factor exposures on disease incidence only (and not mortality) results in a lower prevalence of major illness than if there is an effect on mortality and a smaller overall projected population size compared to our main results.

Our study has some limitations. We modelled the direct effect of risk factors on disease incidence where high-quality, robust evidence was available. However, epidemiological studies may oversimplify the complex interplay between risk factors and diseases. We did not model

causal associations of other factors (e.g. sugar intake or alcohol consumption) for which measurement was unavailable or was inconsistent in the HSE; there are, therefore, possible additional benefits to be gained by tackling these risk factors. For example, we did not model exposure to air pollution. However, the impact of air pollution is small compared to the other risk factors modelled[30]. For conditions where we did not model causal associations with behavioural risk factors (e.g. chronic pain or anxiety and depression), we projected these conditions based on sociodemographic characteristics and indirect relationships modelled between conditions (Supplementary Methods, p15). We have not modelled any behavioural changes resulting from the diagnosis of conditions. We used exposure data from HSE waves up to 2014 and healthcare data up to 2019—periods chosen based on the availability of harmonised data; however, there have been changes in more recent years to both overall trends and socioeconomic inequalities, and the data is from before the coronavirus pandemic. We hypothesise that the higher mortality rates during the pandemic may have affected demographics in England, disproportionately affecting older, multimorbid and frail individuals. However, the reduction in non-COVID-19 related care during

and after the pandemic may have delayed diagnosis and treatment of diseases, leading to less favourable outcomes.

Our analyses measure only diagnosed cases of conditions and do not capture unmet needs. Whilst this linked primary care data largely represents the population of England registered with primary care providers, marginalised groups and those from the most deprived areas are underrepresented. In addition, we assumed that the characteristics of future migrant populations are the same as those within our primary care dataset from 2008 to 2019. Finally, our scenarios do not include time for the implementation of interventions to reduce risk factor exposures. Therefore, our estimates may be optimistic in terms of the time horizon for potential population health benefits.

Our findings suggest that improving risk factor exposure levels will reduce the overall prevalence of major illness within the population compared to the baseline continuing trends scenario. Still, the impact will take several decades to be fully realised. In the short term, lower incidence and mortality rates will still help reduce disease prevalence, improve population well-being, and reduce health and care demand. Benefits are also likely to be seen from delaying the age of onset of chronic conditions that are large drivers of economic inactivity due to long-term illness[31], for example, postponing coronary heart disease or diabetes incidence through reductions in BMI. In other

words, the mixture and severity of conditions among people with major illness might change as a result of the improvements in risk factor exposures, but the Cambridge Multimorbidity Score and our definition of major illness are insensitive to these changes.

Population growth and an ageing population structure will likely remain strong drivers of the future burden of ill health in England. The modelled decrease in mortality rates under our scenarios means that people would likely live for longer, even if this is with poor health, contributing to the relatively small impact of our scenarios on the overall prevalence of major illness in the mid-term. This mirrors findings from a global study projecting alternative rates of change from the Global Burden of Disease studies[16], and a microsimulation of the population-level effects of meeting WHO voluntary targets for six risk factors within Europe[17]. Any public health policy approach will need to adapt to these changing needs, supporting people to improve their risk factor exposure as well as creating places that help prevent exposure to risk factors in the first place.

Our study suggests that some of the biggest gains to population health, as measured by the prevalence of major illness, could be through improving the population-level exposures of BMI and SBP. Eliminating smoking and improving physical activity levels are likely to have a substantial impact on reducing the incidence of major illness, but reduced case fatality under these scenarios may lead to an increased prevalence of major illness. Two multi-country modelling studies on the impact of risk factors on all-cause and cause-specific mortality similarly identified smoking, blood pressure, and BMI as priority targets with the greatest potential for improving life expectancy at a global level[14,16]. Kontis et al. projected smaller potential effects of improved smoking scenarios within high-income countries, where smoking rates have been decreasing over time[14].

The equity impacts of improving risk factor exposure levels relate to the underlying socioeconomic gradients of each individual risk factor, as well as the relationships between the risk factor and incidence and mortality. Of the risk factors modelled, reduction of BMI would likely lead to the biggest absolute reductions in major illness prevalence among those living in the most deprived IMD quintile group, with the opposite pattern for SBP. There is a consistent socioeconomic gradient in projected mean BMI throughout 2023–2043 (Supplementary Results Fig. 1.1). Therefore, a reduction in exposure leads to a greater absolute improvement for individuals living in the most deprived quintile group compared to the least deprived quintile group and impacts a bigger proportion of those in the most deprived quintile group. By contrast, for SBP, in the continuing trends scenario, the mean SBP is highest but projected to decrease fastest among

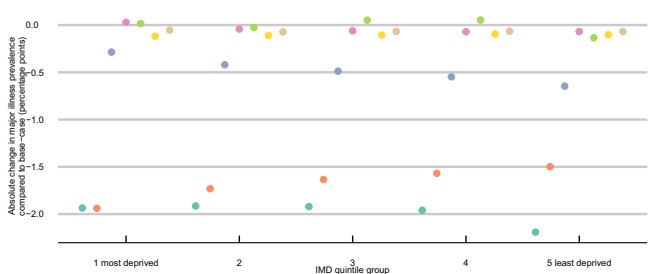

**Fig. 6 | Absolute change in 2043 projected prevalence of major illness by quintile groups of IMD\* by scenario under theoretical minimum risk factor exposure levels.**

● All  ● BMI  ● SBP  ● Smoking  ● Physical activity  ● Fruit & veg  ● Total cholesterol

\*Quintile groups of IMD (Index of Multiple Deprivation) are presented for readability; Major illness is defined as a Cambridge Multimorbidity Score >1.5; SBP systolic blood pressure, BMI body mass index. Source data are provided as a Source data file.

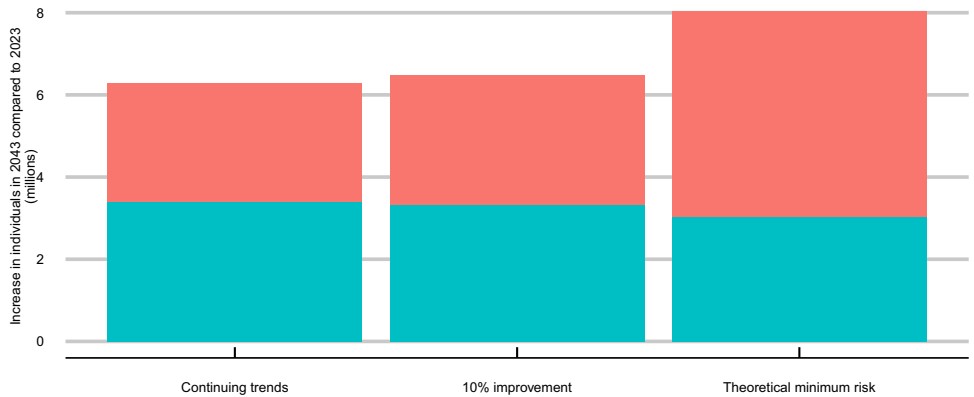

**Fig. 7 | Increase in numbers living with and without major illness in 2043 compared to 2023.**

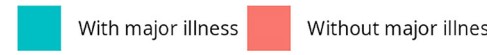
■ With major illness  ■ Without major illness

\*10% improvement and theoretical minimum risk scenario results are presented for the 'all risk factors' scenarios only. Source data are provided as a Source data file.

people in the least deprived quintile group. Furthermore, SBP is primarily linked to CVD, which is one of the main causes of death in the population. Therefore, our SBP scenarios reduce CVD mortality and allow people with major illness to live longer, predominantly in more deprived groups. In addition, the effect of risk improvements impacts age groups differently (Supplementary Results Fig. 5.1), and inequalities in life expectancy are well-documented. For example, a 10% improvement in SBP would have the biggest absolute reduction in major illness prevalence among those aged 70–79 who are more likely to reside in less deprived areas.

The impact on population health of improving risk factor exposure levels within the population by 10% offers an insight into the potential impact of coordinated public health policy action in the UK. Our scenarios are illustrative but are not dissimilar to the impact of previous UK public health policies on SBP and tobacco exposure. For tobacco, 10 years after the introduction of smoke-free legislation in the UK, the prevalence of smoking fell from 21% in 2007 to 15.5% in 2017, a 27% relative reduction[32], and then a further 17% relative reduction to 12.9% in 2022[33]. Analysis of the salt intake reduction programme in England introduced in 2003 reported a reduction in average SBP from 125.3 mmHg in 2003 to 122.6 mmHg in 2014 (20.3% relative reduction in excess risk), which then plateaued from 2014 to 2018 (122.04 mmHg)[34].

In contrast, the complex, multifaceted nature of obesity means that similar reductions in obesity prevalence have not been achieved. The UK Soft Drinks Industry Levy led to a reduced average weekly consumption of sugar by 3.3 grams in the year after implementation (April 2018)[35], and is estimated to have led to an 8% relative reduction in obesity rates among age 11 girls[36], but its impact on obesity reduction among the adult population is estimated to be far smaller[37]. In addition, a 9 pm watershed on the advertising of food and beverages high in fat, sugar, and salt is projected to reduce the prevalence of childhood overweight and obesity by 3.6%, an absolute reduction of 1.2 percentage points[38]. Whilst implementation of this legislation was initially planned for 2022, it has been delayed until 2025[39], illustrating the difficulty in introducing effective, evidence-based policies. A broader cross-government approach to prevention and inequalities will be needed, therefore focusing not just on direct causes of obesity, such as unhealthy food, but also on the wider determinants of health, such as green space, quality housing and a good education[40]. Future work could explore the absolute and equity impact of specific policy options, incorporating healthcare costs and quality-adjusted life years to estimate the cost-effectiveness of policy interventions. It would also be informative to expand to modelling other risk factors, depending on the availability of sufficient data, such as alcohol, salt, and air quality.

In summary, our scenarios highlight how improving risk factor exposure levels within the English population can reduce the future incidence of major illness. However, reduced case fatality rates alongside an ageing population mean the total number of people living with major illness will likely increase, and therefore, demand for healthcare services and management of complex conditions will likely continue to grow. Substantive action is therefore needed for primary prevention efforts, support for healthy ageing, and the management of ill health. In addition to targeted action on risk factors, policymakers will need to engage in a much more coherent cross-government approach to disease prevention, focusing on the wider determinants of health alongside adapting health and care systems to manage an increasingly multimorbid population.

## Methods
### Microsimulation overview
We used IMPACT$_{NCD}$, a validated dynamic discrete-time microsimulation model, to simulate the effect of varying levels of reduction in selected risk factors on future trends in major illness. IMPACT$_{NCD}$ simulates the life course of synthetic individuals aged 30 and over from a close-to-reality English population and translates changes in risk factor exposure trends into changes in annual disease incidence, case fatality, and disease prevalence. Within IMPACT$_{NCD}$, each unit is a synthetic individual (simulant) represented by a record containing a unique identifier and a set of associated attributes. The microsimulation then projects the life course of each synthetic individual. The attributes of each synthetic individual include sociodemographic characteristics, exposures to risk factors, acquired diseases, and the cause of death if relevant. All these attributes are updated in discrete annual steps according to a set of stochastic rules. We structured these rules based on well-established epidemiological principles. Specifically, behavioural risk exposures are conditional on sociodemographic exposures; biological risk exposures are conditional on behavioural and sociodemographic exposures, and diseases are conditional on biological, behavioural, and sociodemographic exposures, as well as the diagnosis of other conditions. Finally, mortality depends on sociodemographic, behavioural, biological and disease exposures. The technical appendix (Supplementary Methods) provides a detailed description of the microsimulation model; key assumptions are summarised in Table 1; model source code is available here: https://zenodo.org/records/17085041. IMPACT$_{NCD}$ has been used extensively to model primary prevention policies nationally in England, Germany, Brazil, and the US, and locally in Liverpool[23–27,41–46].

### Data sources
This iteration of IMPACT$_{NCD}$ uses data from adults aged 20 and over from the HSE 2003–2014 to inform trends in risk factors ($N = 113,780$)[47]. Linked primary care records (CPRD Aurum linked to Hospital Episodes Statistics (HES) inpatient, HES outpatient, and ONS mortality records) of 1.7 million adults aged 20 and over from 2008 to 2019 are used to inform trends in disease incidence, prevalence, and disease-specific mortality[48]. ONS population estimates[49,50] and projections (2021-based)[29] are used to inform estimates of the population size and structure and to calibrate mortality.

### Risk factors and disease incidence
We included three sociodemographic risk factors of sex, age, and decile of the Index of Multiple Deprivation (DIMD), plus eight amenable risk factors: BMI, smoking status and history, SBP, fruit consumption, vegetable consumption, total cholesterol, physical activity, and environmental tobacco smoke. The direct causal relationships between amenable risk factors and disease incidence included in the model were chosen based on the availability of high-quality evidence from systematic reviews and meta-analyses and are displayed in Table 2. For some conditions, we also modelled prevalent diseases as risk factors for other diseases (i.e. diabetes mellitus is a risk factor for coronary heart disease) using relative risks derived from our analyses of linked primary care data (see Supplementary Methods). In addition to the causal associations between risk factors and disease incidence, we modelled causal associations between risk factors and disease-specific case fatality, assuming the same relative risks. Furthermore, we allowed risk factors (i.e. SBP, smoking, physical activity, and several diseases) to influence mortality from causes of death not explicitly modelled. We conducted a sensitivity analysis where improved risk factor exposures did not impact case fatality.

### Health outcomes
The Cambridge Multimorbidity Score (CMS) is a composite index of 20 long-term conditions that comprise 65% of disability-adjusted life years in England[7], with each condition weighted according to its impact on the health system and an individual's health outcomes[28]. So that we could explicitly model certain causal relationships, we modelled 26 conditions that we then aggregated to the 20 CMS conditions; for example, we modelled Type 1 and Type 2 Diabetes separately and

## Table 1 | IMPACT$_{NCD}$ key assumptions and limitations

| Model component | Key assumptions |
|---|---|
| Sociodemographic module | Migration is not modelled explicitly in the model. However, the model outputs are calibrated to ONS population projections, which include migration (we used the ONS principal migration assumptions). Nevertheless, we assume that migrants have similar characteristics to the local population. |
| | Social mobility is not considered. |
| | Decile groups of the index of multiple deprivation (DIMD) are a relative marker of (area) deprivation with several versions since 2003. We have used the 2015 version and assume it is constant throughout the simulation[54]. |
| Exposure module (see Supplementary Methods Exposure module section p7). | We assume that the surveys used are truly representative of the population. For example, the adjustments for selection bias in the Health Survey for England are adequate. |
| | On average, simulants remain in the same exposure quintile group throughout their life. |
| | The linear correlations in exposure quintile groups remain constant over time (i.e. the clustering of exposures in some subpopulations). |
| | We assume that trends in risk factor exposures continue and follow log-linear trends. |
| Disease module | We assume multiplicative risk effects (see Supplementary Methods Disease Incidence section p10). |
| | We assume a log-linear exposure-response for the continuous risk factors. |
| | We assume that the effects of the risk factors on incidence and case fatality are equal (see Supplementary Methods Mortality section p16). |
| | We assume a mean lag time between exposure and outcome of about 4–5 years for most exposure/outcome pairs, except for cancers, for which we assume a mean lag time of 9 years (see Supplementary Methods Disease Incidence section p10). |
| | We assume 100% risk reversibility for all exposures except smoking. We allow smoking to have a cumulative effect on the risk for COPD and lung, breast, and colorectal cancers. |
| | We assume that trends in disease incidence are attributable only to trends of the relevant modelled risk factors or other diseases modelled. |
| | We assume that the linked primary care data used to model disease trends over time represents England's adult population. |
| | We assume that trends in disease incidence continue to follow log-linear trends (other than pain, see below). |
| | For cancers, we assume that survival 10 years after diagnosis equals remission. |
| | For all conditions other than cancer (see point above), pain, constipation, asthma, alcohol problems, and anxiety and depression, we assume conditions are chronic. |
| | For pain, we modelled the incidence of pain based on the incidence in 2013 due to data quality issues over time with the prescription data. |
| | For anxiety and depression and constipation, we did not calibrate to the observed trends in incidence rates because their projections led to implausible rates. |

then combined them into Diabetes (Table 2). Aligned with our previous publications[1,51], we used a CMS score greater than 1.5 to denote 'major illness' and tested this threshold as a sensitivity analysis.

We present the absolute and relative difference in the prevalence of major illness within the population (aged 30 and over) compared to a base-case scenario continuing the observed trends in risk factors from 2003 to 2014. We also present the effect of risk factor improvements on all-cause mortality and incidence rates, the number of case-years lived with major illness prevented/postponed compared to the continuing trends scenario, and the change in the number of people living without major illness.

We simulated health outcomes from 2013 to 2043, and the results are presented from 2023, the year the scenario was implemented. We used the period before 2023 for validation and calibration.

## Scenarios

As our base-case scenario, we used projected continuing trends in exposure to risk factors, disease burden, and mortality. For each of the risk factors, we considered two levels of improvement compared to the continuation of recent trends:

1. Theoretical minimum risk: eliminating the excess risk to estimate the proportion of major illness burden attributable to each risk factor.
2. A 10% relative improvement to the base-case exposure level in each year.

Table 3 summarises the implementation of each of the scenarios. We combined fruit and vegetables into one scenario; environmental tobacco smoking is calculated from smoking prevalence. The risk factor changes occur in 2023 and continue throughout the simulation period. For each level of improvement, we reduced each of the risk factors individually (holding all others constant) and then reduced all risk factors simultaneously.

Theoretical minimal risk levels were defined based on those used in the Global Burden of Disease studies[7]. The 10% improvement was calculated each year relative to the base-case exposure in that year. For biological risk factors where the theoretical minimum risk level is not 0 (BMI, total cholesterol, SBP), the 10% improvement was calculated as a 10% decrease in the excess risk compared to the theoretical minimum risk. For smoking, the number of smokers was reduced by 10%, and among smokers, the number of cigarettes smoked per week was reduced by 10%. For physical activity, we increased the number of days of physical activity per week by one for 10% of the population in the continuing trends scenario. Improvement in fruit and vegetable consumption was implemented as a 10% increase in grams daily. For the 10% improvement in BMI scenario, we modelled the indirect effect of decreasing BMI on SBP and total cholesterol: for BMI ≥ 25, for each 1 unit decrease in BMI, we modelled a 0.23 mmol/L (95% confidence interval (CI): 0.15, 0.30) decrease in total cholesterol[52], and a 2.55 mmHg (95% CI: 1.55, 3.54) decrease in SBP[53]. As a sensitivity analysis, we also modelled a direct effect-only BMI scenario without these indirect effects on SBP and total cholesterol.

**Table 2 | Overview of how individual conditions are modelled**

| CMS condition | CMS score[a] | Modelled condition | Recovery and recurrence | Direct causal relationships | | Pre-existing conditions[b] |
|---|---|---|---|---|---|---|
| | | | | Risk factors | | |
| Dementia | 2.50 | Dementia | No recovery | BMI, smoking | | Y |
| Cancer | 1.53 | Breast cancer | Recovery after 10 years of survival; no recurrence | Physical activity, BMI, smoking, ETS | | Y |
| | | Colorectal cancer | | Physical activity, BMI, smoking | | Y |
| | | Lung cancer | | Smoking, ETS, fruit consumption | | Y |
| | | Prostate cancer | | Smoking | | / |
| | | Other cancers | Recovery after 10 years of survival; can recur | / | | Y |
| COPD | 1.46 | COPD | No recovery | Smoking, ETS | | / |
| Atrial fibrillation | 1.34 | Atrial fibrillation | No recovery | BMI, smoking, SBP | | Y |
| Heart failure | 1.18 | Heart failure | No recovery | / | | Y |
| Constipation | 1.12 | Constipation | Recovery is stochastic; can recur | / | | Y + past constipation |
| Epilepsy | 0.92 | Epilepsy | No recovery | / | | Y |
| Chronic pain | 0.92 | Chronic pain | Recovery is stochastic; can recur | / | | Y + past chronic pain |
| Stroke/transient ischaemic attack | 0.80 | Stroke/transient ischaemic attack | No recovery | Physical activity, BMI, Smoking, ETS, SBP, fruit and vegetable consumption, total cholesterol | | Y |
| Diabetes (type I or II) | 0.75 | Diabetes Type 1 | No recovery | / | | / |
| | | Diabetes Type 2 | No recovery | Physical activity, BMI, Smoking, ETS, SBP, fruit consumption | | / |
| Alcohol problems | 0.65 | Alcohol problems | Recovery is stochastic; can recur | / | | Past alcohol problems |
| Psychosis/ bipolar disorder | 0.64 | Psychosis/bipolar disorder | No recovery | / | | Y |
| Chronic kidney disease | 0.53 | Chronic kidney disease | No recovery | BMI, SBP | | / |
| Anxiety and depression | 0.50 | Anxiety and depression | Recovery is stochastic; can recur | / | | Y + past anxiety and depression |
| Coronary heart disease | 0.49 | Coronary heart disease | No recovery | Physical activity, BMI, Smoking, ETS, SBP, fruit and vegetable consumption, total cholesterol | | Y |
| Connective tissue disorders | 0.43 | Rheumatoid arthritis | No recovery | / | | / |
| | | Other connective tissue disorders | No recovery | / | | / |
| Irritable bowel syndrome | 0.21 | Irritable bowel syndrome | No recovery | / | | / |
| Asthma | 0.19 | Asthma | Recovery is stochastic; can recur | BMI, smoking | | Past asthma |
| Hearing loss | 0.09 | Hearing loss | No recovery | / | | / |
| Hypertension | 0.08 | Hypertension | No recovery | SBP | | / |

CMS Cambridge multimorbidity score, ETS Environmental Tobacco Smoke, SBP Systolic blood pressure, COPD Chronic Obstructive Pulmonary Disease.
[a]CMS score: general outcome weights[28].
[b]Supplementary Methods Fig. 2.2 shows the size and direction of modelled associations between conditions.

**Table 3 | Summary of implementation of risk factor improvement scenarios compared to our base-case scenario of projected continuing trends in exposure to risk factors, disease burden, and mortality**

| Risk factor | Model variable | 10% improvement in risk factor exposure[a] | | Theoretical minimum risk level of exposure (all individuals affected) |
|---|---|---|---|---|
| | | Who in the base-case population is affected | 10% improvement | 10% improvement |
| Smoking | Current smokers stop smoking | Smokers | % increase of base-case | All smokers become ex-smokers and smoke initiation rate set to 0 |
| | Number of cigarettes smoked per day | Smokers | % decrease of base-case | Set to 0 |
| | Environmental tobacco smoking | Non-smokers | Calculated in the model from the above | No environmental tobacco smoking |
| Fruit and vegetables | g/day of fruit + g/day of vegetables | All | % increase of base-case | Theoretical minimum risk: 4 portions of each[b] |
| Physical activity | # of active days | 10% of anyone with <7 days of physical activity per week | An increase of 1 day of physical activity per week | 7 days of physical activity |
| BMI[c] | BMI (kg/m2) | All | 10% decrease of excess risk compared to theoretical minimum risk[b] | Theoretical minimum risk: BMI = 22[b] |
| Total Cholesterol | Total cholesterol (mmol/L) | All | 10% decrease of excess risk compared to theoretical minimum risk[b] | Theoretical minimum risk: Total cholesterol = 4 mmol/L[b] |
| SBP | SBP (mmHg) | All | 10% decrease of excess risk compared to theoretical minimum risk[b] | Theoretical minimum risk: SBP = 112[b] |

BMI Body mass index, SBP Systolic blood pressure.

[a]The 10% improvement is calculated annually relative to the base-case scenario for that year, e.g. in 2023—10% of base-case in 2023, in 2024—10% of base-case in 2024.

[b]Theoretical minimal risk exposure levels are derived from a distribution based on high-quality studies [see Supplementary Methods page 34–136] and, therefore, differ in each model iteration; presented values are the mean for the 100 iterations of this model. An illustrative example with BMI: if the BMI of individual X in the base-case scenario is 28, in a 10% improvement scenario, this would be 27.4 [(28–22) × 0.1 = 0.6 reduction in BMI].

[c]We also model the mediating effect of decreasing BMI on SBP and total cholesterol.

## Model validation and calibration

We validated the IMPACT$_{NCD}$ epidemiological engine using internal validation, plotting the modelled exposures' prevalence and disease incidence against the observed exposures' prevalence and disease incidence in HSE and linked primary care data, respectively. Mortality in the model is calibrated to ONS mortality projections (see Supplementary Methods−Mortality Calibration section (p16)). Our risk factor projections follow similar patterns to observed values from HSE stratified by year and age group and by quintile groups of IMD and age group (Supplementary Methods p137−175). For our primary outcome, prevalence of CMS > 1.5 (defined as 'major illness' in this paper), we plotted projected major illness against the observed CPRD Aurum prevalence (see Supplementary Methods Fig. 5.1 (p19)). Validation plots for incidence, case fatality, and prevalence of individual conditions are shown in the Supplementary Methods (p40−132). Overall, the validation plots suggest that IMPACT$_{NCD}$ captures exposure trends and translates them to disease incidence and mortality reasonably well for the purpose of this project.

## Ethical approval

The Clinical Practice Research Datalink (CPRD) Independent Scientific Advisory Panel approved the study protocol [ISAAC 20_000096]. CPRD has ethics approval from the Health Research Authority to support research using anonymised patient data.

## Reporting summary

Further information on research design is available in the Nature Portfolio Reporting Summary linked to this article.

## Data availability

This study did not generate any original raw data. All data used were from secondary data sources, each of which are detailed below. *Linked CPRD-HES-ONS data*: The individual-level health data used in this study were obtained from the Clinical Practice Research Datalink (CPRD) Aurum database after approval by the CPRD independent scientific advisory panel (protocol ISAAC 20_000096). A detailed protocol for this study can be provided upon request. Access to anonymised data from CPRD is subject to a full licence agreement containing detailed terms and conditions of use. Anonymised patient datasets can be extracted for researchers against specific study specifications, following protocol approval. Further information is available at www.cprd.com/data-access. Data citation: Clinical Practice Research Datalink. (2021). CPRD Aurum June 2021 (Version 2021.06.001) [Dataset]. Clinical Practice Research Datalink. https://doi.org/10.48329/pyc2-we97 *HSE data*: Data on risk factor exposures were obtained from the Health Survey for England, made available via the UK Data Service (UKDS). Access to UKDS data is subject to their end-user licence agreement. Data citation: NatCen Social Research, University College London, Department of Epidemiology and Public Health. (2024). *Health Survey for England*. [data series]. *8th Release*. UK Data Service. SN: 2000021, https://doi.org/10.5255/UKDA-Series-2000021. *ONS data*: Data on population estimates, population projections, and mortality are available directly from the Office for National Statistics website. Data citation: Office for National Statistics (ONS), released 30 January 2024, ONS website, statistical bulletin, National population projections: 2021-based interim. Available from: https://www.ons.gov.uk/peoplepopulationandcommunity/populationandmigration/populationprojections/bulletins/nationalpopulationprojections/2021basedinterim. Number of deaths and populations in deprivation decile areas by sex and single year of age, England and Wales, registered years 2001 to 2018−Office for National Statistics [Internet]. [cited 2022 Feb 18]. Available from: https://www.ons.gov.uk/peoplepopulationandcommunity/birthsdeathsandmarriages/deaths/adhocs/11169deathregistrationsandpopulationsbyindexofmultiplede privationengland2001to2018. *Derived data*: Aggregated data

underlying the microsimulation model is available here: https://zenodo.org/records/17085041. Summary tables derived from the model output for this study are available here: https://zenodo.org/uploads/17084698. Source data for all figures are provided with this paper in the 'Source data.xlslx' file Source data are provided with this paper.

## Code availability

Code for the model is available here: https://zenodo.org/records/17085041.

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

## Acknowledgements

This work is based in part on data from the Clinical Practice Research Datalink obtained under licence from the UK Medicines and Healthcare products Regulatory Agency. The data are provided by patients and collected by the NHS as part of their care and support. The interpretation and conclusions contained in this study are those of the authors alone. Linked HES and ONS data Copyright © (2021) were reused with the permission of The Health & Social Care Information Centre. All rights reserved. This work was funded by The Health Foundation (C.K., M.O.F., B.C., A.H., M.B.). This work is the result of a partnership between researchers at the Health Foundation (the funder) and researchers at the University of Liverpool. The Health Foundation is an independent charitable foundation whose mission is to improve health and care in the UK.

## Author contributions

C.K., T.W. and A.H. developed the original idea and designed the study and analysis plan. T.W. and C.K. acquired the data. A.H. and C.K. directly accessed and verified the underlying data. Analysis was done by A.H., and reviewed by C.K. A.H. wrote the first draft. All authors (A.H., A.R., L.R.J., A.B., B.C., M.B., A.C., M.O.F., T.W., C.K.) contributed to interpreting the results and critically revising the draft, and all agreed on the final version. All authors approved the final draft, had full access to all data in the study, and accept responsibility to submit for publication.

## Software
The IMPACTNCD_Engl model is written primarily in the R programming language, for which version 4.2.3 was the latest release at the time of submission. C++ code (ISO/IEC 14882 standard) was also used.

## Competing interests
No authors have declared conflicts of interest. This study was funded by the Health Foundation; authors T.W., A.R., L.R.J., A.C. and A.B. were employed by the Health Foundation when conducting this work.
