## [Transparent Peer Review file · Nature Communications]

Exploring the contribution of risk factors on major illness: a microsimulation study in England, 2023-2043

Corresponding Author: Dr Anna Head

Version 0:

Reviewer comments:

Reviewer #1

(Remarks to the Author)

The manuscript uses a microsimulation model to project trends in prevalence rate, incidence rate, case-years prevented or postponed and numbers living without major illness, as well as trends in mortality for years 2023-2043 among the population of England aged 30 and over under three scenarios of the levels of modifiable risk factor. The biggest contributors to changes in major illness were body mass index, systolic blood pressure, smoking and physical activity. However, improving the risk factor levels lowered the disease burden only little because of the aging of the population. As a conclusion, despite the expected incidence of major illness due to improving risk factor exposure, the demand for health care services is likely to grow. This is caused by the aging of the population structure and the fact that people with disease will survive longer.

I have reviewed the paper mainly from epidemiological and statistical perspectives, and I have less expertise in clinical and health service sectors.

Projection of chronic conditions is challenging because of complex interrelations between and among the diseases and their determinants and limited availability of good data. The authors have made use of the relatively good national data sources that are available in England. They have also incorporated, as far as I can see, the best available knowledge about the relationships between the components, justified their assumptions on issues where data are lacking, and documented thoroughly the simulation model and its validation in the Technical Appendix. They have also made sensitivity analyses of various critical issues and acknowledged relevant limitations. As the result, I find the results as valid as is reasonable based on the available data. I will specify below any comments on the details. The interpretation and conclusions of the results are justifiable.

The conclusions are important for those planning and deciding on preventive actions, support for healthy ageing and health care systems. Although the results are specific to England, the results are also relevant to other countries. In fact, only few countries have reasonable national data sources needed for such projections.

Here are my comments and suggestions:

1. The two supplements (a) Technical appendix (which in some places is apparently called Supplementary materials), and (b) Supplementary results should be labelled and referenced clearly and consistently. Now, for example, it is not always clear in which of the two supplements to search for Tables B.
2. The description of the microsimulation model in the online methods requires clarification. In fact, I largely misunderstood the paper after first reading, until I read the very clear description of the methods in the Technical appendix. It may be sufficient to add to the online methods the specification that that it is a discrete time model which simulates individuals' yearly changes in risk factors, health and survival. (The alternative that confused me, when reading the paper first time, was a continuous time model which would simulate the times of change of individuals' status.)
3. The base-case scenario of continuing trend has a key role in the results. Therefore, its selection should be justified and discussed. The base-case trend is based on data from much before the projection period, and there is not reason to assume that the risk factor trend would stay linear for forty years. Changes are possible for both directions. Nevertheless, I think the current selection is good because there are no justifiable alternatives. However, if the simulation model allows, I suggest to

add a fourth scenario of no trend in risk factors. This would have a clear interpretation and would give an informative comparison in addition to the current scenario of theoretical minimum risk.

4. Setting single risk factors in turn to the theoretical minimum and assuming that all other risk factors follow the base-case trend is not a very realistic scenario because, in particular, BMI is known to affect SBP. I suggest that, in the main results, the authors consider replacing this assumption with the more plausible assumption which allows SPB to change with BMI. If I understand it correctly, such projections have already been made, and are now reported as "BMI sensitivity" in some of the supplementary tables. This change would make the sum of the contributions of the individual risk factors larger than the contribution of all risk factors together, but this is the reality because of the associations between the risk factors.

5. A major determinant of the results is the changing age structure of the English population. It would be helpful for the reader to add figures of the (observed or projected) age structure in selected years (e.g. 2023, 2033 and 2044). This could go to the Supplementary results.

Specific comments (the line numbers refer to the pdf version of the documents provided to the reviewers):

6. Introduction: Where should references 8-10 go, if relevant?

7. Lines 75 and 541: Environmental tobacco smoke (ETS) is not mentioned elsewhere in the manuscript, and it is only in the tables of risk factor trends in the Supplementary results. Was its impact really estimated in the analysis? If it was, the results should be reported.

8. Lines 85-86: Figures A.5-A.6 of the Technical appendix do not relate to this. I guess the authors mean Tables B.1-B.2 of the Supplementary results. However, Table B.2 is stratified by age groups, not by birth cohorts.

9. Line 92 and throughout the manuscript: The meaning of the numbers in brackets should be explained? I guess they show the "uncertainty intervals" defined on page 17 of the Technical appendix, which cover some of the uncertainty of the projections.

10. Line 150: Correct "Figure 5" to "Figure 2".

11. Line 174 and elsewhere: The four quintiles divide the distribution into five parts, usually called the "fifths". Figure 6 clearly shows results by the fifths rather than on the quintiles.

12. Lines 175-180: I suspect over-interpretation of the precision of the projections. The differences are so small that I would rather say "similar" instead of "greater" or "mixed".

13. Line 221: If sensitivity analysis 1) is the same as described on line 552, the latter description is much clearer.

14. Lines 244-245: Specifying such details before describing the overall structure of the model is confusing. Adding a reference to the online methods or the technical appendix should be sufficient here.

15. Line 258: Substitute "the Health Survey for England (HSE)" for "HSE".

16. Line 312: The missing reference is apparently Supplementary Figure B.1.

17. Line 317: The missing reference is apparently Supplementary Figure B.2.

18. Line 524: The reference is apparently Table 2-1 of the Technical appendix.

19. Line 525: I failed to find model source code in reference 33.

20. Line 528: Shouldn't the reference be 33, not 34,

21. Line 523: Shouldn't the references be "34, 35, 36".

22. Line 544: I guess the reference should be to the Technical appendix.

23. Line 549: By "survival" you apparently refer to case-fatality, which is defined in the Technical appendix. Consistent terminology would help the reader.

24. Lines 564-565: I find this sentence confusing in the description of methods. This is indicated neither in the title of the paper nor in the objectives in the Introduction.

25. Line 566: I suggest to clarify as "... continuing the trends of 2003-2014 in risk factors."

26. Line 580: Add a reference to the Technical appendix.

27. Line 586: I suggest to specify the minimal levels here and add a reference to page 13 of the Technical annex.

Figures: I failed to see some of the light-coloured lines on the figures.

(Remarks on code availability)

As mentioned in my comments to the authors, I failed to find the code in the URL given in the manuscript. I now found the code in your URL. Although I don't have the capacity to review the code, I am happy to see that it is publicly available.

Reviewer #2

(Remarks to the Author)

The authors present a variety of scenarios in terms of a multiple changes in the prevalence of behavioural and other risk factors on morbidity. The objective is a worthy one but I found the execution in this paper unconvincing.

Structure:

The paper goes from an introduction straight into results. What happened to a methodology section?

Writing style:

I found the introduction section to be quite tersely written and not an easy read. Even though the content was fine, it lacked flow.

Software:

The results are based on the authors' own IMPACT software tool. This is perhaps fine, but I could not see any reference to the model underpinning the software having been independently peer reviewed in some way.

As a result, when you launch into the Results section with no build up, there is no context for the projected exposure levels. Where did these come from? Where is the historical data to set these up?

And there is no explanation that I could easily see on how the behavioural changes interact with outcomes.

BMI:

Is mean BMI the sole variable rather than a distribution across all levels of BMI that shifts over time. If it is just the mean then there is a problem with the underlying model. The reason for this is that the risk of developing disease has a non-linear relationship with BMI. So you really need to model separately e.g. the % who are 1: morbidly obese or 2: obese or 3: overweight etc.

The same might go for other risk factors.

Note that there is difference between BMI which is effectively an intermediate risk factor whereas other risk factors such as smoking, fruit and veg, and activity are behavioural risk factors more in the control of the individual.

Other:

On page 6 and several other places there is a very obvious (i.e. in bold) "Error!". This shows that the authors have not checked the version submitted and is somewhat disrespectful towards the journal and referees.

(Remarks on code availability)

Reviewer #3

(Remarks to the Author)

This paper utilizes the validated microsimulation model (IMPACTNCD) to project the potential effects of improving population-level exposure to eight risk factors on the burden of major illness among adults aged 30 and older in England from 2023 to 2043. While the study offers valuable insights into how risk factor improvements could shape future health outcomes, several limitations and areas requiring clarification remain, which could enhance the methodological rigor and interpretability of the findings:

1. The selection of eight risk factors is not fully justified. Important contributors to chronic diseases, such as alcohol consumption and air pollution, are excluded, raising concerns about the completeness of the projections. The authors should explain the rationale for focusing on these specific risk factors. Whether the choice was influenced by data limitations in the Health Survey for England (HSE)?
2. The manuscript does not explicitly address whether interactions or mediating effects between risk factors (e.g., BMI influencing total cholesterol or systolic blood pressure) were considered in the model. Given the complexity of chronic disease etiology, such interactions are critical. The authors should clarify whether these effects were incorporated, describe how they were modeled if applicable, and discuss the implications for the projections if they were not.
3. It is unclear whether the 26 diseases modeled correspond to all the long-term conditions listed in the Cambridge Multimorbidity Score (CMS). The authors should clarify this alignment and provide specific data or a comparison to substantiate their selection. Additionally, the advantages of focusing on these 26 diseases should be discussed.
4. Although the manuscript mentions that the IMPACTNCD model is well validated, the main text lacks details on the validation process and results. This reliance on supplementary materials makes it difficult for readers to access critical information. Including a concise summary of key validation outcomes, such as predictive accuracy, error margins, or comparisons with other models, in the main text would improve the transparency and credibility of the study.

(Remarks on code availability)

Version 1:

Reviewer comments:

Reviewer #1

(Remarks to the Author)

The authors have addressed satisfactorily most of my comments. I only want to comment on the responses to three of my earlier comments:

3. I agree with the response, but it does not address my earlier suggestion. (Perhaps my comment was unclear). I was not questioning the use of continuing trends as the baseline scenario. I suggested to add the interesting scenario of no trend. I think is even more realistic than the scenario of theoretical minimum risk, which is already included in the paper. I hope the authors consider this suggestion, but I don't insist on it.

According to the response, the authors have now validated the projections of risk factor trends against a more recent health survey for England data (2015-2019). Why not mention this valuable information on the paper?

12. The authors' response highlights the importance of my concern but does not remove the problem: The reported differences in percentage points correspond to large numbers of persons with major illness, but comparison of the provided uncertainty intervals does not convince the reader about real differences. The authors should recognize this fact in the paper.

It is possible that the comparison of the IMD groups is actually more precise than suggested by the reported uncertainty intervals. If this is the case, it should be stated in the paper.

23. Third paragraph of the Discussion: Shouldn't some of the words "mortality" be replaced with "case fatality"?

(Remarks on code availability)

Reviewer #3

(Remarks to the Author)

Thank you for your responses. Regarding point 2, I appreciate the clarification on BMI's indirect effects. However, my concern extends beyond this specific example. Chronic disease risk factors often interact in complex ways, which could influence disease progression and overall projections. Could the authors clarify whether additional interactions or mediating effects were considered in the model? If these were not explicitly accounted for, a brief discussion of this limitation and its potential impact on the findings would be valuable.

(Remarks on code availability)

Reviewer #4

(Remarks to the Author)

I thank the authors for their concerted efforts in addressing the comments provided by all three reviewers. In particular I note that the Introduction is substantially improved, and now provides a readable yet compact examination of this research area and methodology.

I also commend the authors' work on the Discussion, which delineates very clearly the limitations of this study and the data used to inform the model, and concludes with a cogent analysis of the difficulties of developing and implementing evidence-based policy for the reduction of major illness. My two immediate concerns with this study -- that the data originated from before the Covid-19 pandemic, and that behavioural changes in response to diagnoses were not modelled -- were appropriately called out in this section.

I feel this study also makes an implicit case for the benefits of applying multiple simulation approaches to major population health challenges. Some of the details lacking here (e.g., the behavioural changes in response to diagnosis, unmet need, etc.), can be captured in agent-based models using appropriate behavioural assumptions vetted by domain experts (which is what I do, incidentally). Conversely, the microsimulation work enhances related complex-systems-based methods like ABM by providing projections and outcomes that generate policy ideas, which can then be tested and examined in the ABM framework. In future work, I feel it would be productive to seek out collaborative relationships with research groups using such methods to cover the gaps in microsimulation, and in turn, to assist the ABM researchers in constructing their models with greater realism and precision, which is often very difficult in ABM work.

The current sensitivity analysis does not provide detailed uncertainty quantification nor does it quantify the level of contribution of different model parameters to the final output variance. More sophisticated techniques like Gaussian process emulation or similar provide these outputs, and are useful for model calibration as well as sensitivity analysis and uncertainty quantification.

Overall, I feel the manuscript is providing a useful examination of major health challenges in England which are illuminating for policymakers and other modellers, and the provision of the code under the GPL licence is very helpful for the modelling community in this area. I believe the authors have adequately modified and clarified the paper to address the concerns of all three reviewers.

I would make only two very small additional requests: firstly, line 142 is missing the 'P' in 'Projections'; and Section 3 of the GitHub repository's documentation should be completed with appropriate references to prior publications using ImpactNCD for the benefit of future users prior to publication of this paper.

(Remarks on code availability)

The code provided in the GitHub repository is provided with detailed instructions for installation including screenshots, so the model is very easy to install. The code appears to run as advertised; however, as I do not use R for simulation modelling but instead general-purpose programming languages like Python and Julia, I cannot evaluate the readability of the code itself.

I do note however that Section 3 of the ReadMe on the repository (Further Notes and References) is not filled out. This should be done so that potential users can refer back to previous published works produced using this modelling framework.

Version 2:

Reviewer comments:

Reviewer #1

(Remarks to the Author)

I am happy with the Authors' response to my earlier comments, except a small terminology issue in the Results. The subsection "Impact on inequalities" uses terms "quintile group", "fifth" and "quantile" for the same issue. To make the paper easier to read, I suggest the authors unify the terminology (throughout the paper). I prefer the term "fifth", but would also be happy with "quintile group" which is a new term to me. "Quantile" (or more specifically "quintile") refers to a point and therefore is incorrect.

(Remarks on code availability)

Reviewer #3

(Remarks to the Author)

The authors have improved the manuscript.

(Remarks on code availability)

Reviewer #4

(Remarks to the Author)

Having read through the authors' rebuttal letter and perusing the updated manuscript and code repository, I'm pleased to say that the authors seem to have taken all my comments into account. I can see that significant text has been added/changed to reflect the comments of the other reviewers as well, which has improved the study by making the limitations of the methods more clear, as well as clarifying some of the terminology used.

From my perspective, the work is a scientifically sound piece of microsimulation research which provides useful insight into the contribution of various risk factors to major illness in England. I feel the study will provide value to policymakers as well as researchers, who will be able to access and built upon the open-source code provided. Given that my comments and concerns were addressed, I would therefore recommend that this version of the manuscript be approved for publication.

(Remarks on code availability)

The code is well-documented and runs well. As someone who uses general-purpose programming languages like Python, Julia and C rather than R, I cannot comment on the readability of the code itself. In my previous review, I noted that the readme on the GitHub repository was missing some valuable context and links to previous works; this has now been rectified.

From my perspective the authors have done a fine job in documenting their code and ensuring that potential users can easily make use of it. Having said that, I personally prefer using more open licensing than the GPL (Creative Commons or the MIT License), given that our work is publicly funded, but the GPL is good enough.

NCOMMS-24-70452-T Response to Reviewer comments

Reviewer #1 (Remarks to the Author):

The manuscript uses a microsimulation model to project trends in prevalence rate, incidence rate, case-years prevented or postponed and numbers living without major illness, as well as trends in mortality for years 2023-2043 among the population of England aged 30 and over under three scenarios of the levels of modifiable risk factor. The biggest contributors to changes in major illness were body mass index, systolic blood pressure, smoking and physical activity. However, improving the risk factor levels lowered the disease burden only little because of the aging of the population. As a conclusion, despite the expected incidence of major illness due to improving risk factor exposure, the demand for health care services is likely to grow. This is caused by the aging of the population structure and the fact that people with disease will survive longer.

I have reviewed the paper mainly from epidemiological and statistical perspectives, and I have less expertise in clinical and health service sectors.

Projection of chronic conditions is challenging because of complex interrelations between and among the diseases and their determinants and limited availability of good data. The authors have made use of the relatively good national data sources that are available in England. They have also incorporated, as far as I can see, the best available knowledge about the relationships between the components, justified their assumptions on issues where data are lacking, and documented thoroughly the simulation model and its validation in the Technical Appendix. They have also made sensitivity analyses of various critical issues and acknowledged relevant limitations. As the result, I find the results as valid as is reasonable based on the available data. I will specify below any comments on the details. The interpretation and conclusions of the results are justifiable.

The conclusions are important for those planning and deciding on preventive actions, support for healthy ageing and health care systems. Although the results are specific to England, the results are also relevant to other countries. In fact, only few countries have reasonable national data sources needed for such projections.

Response: Thank you for your detailed comments and for taking the time to carefully review our paper.

Here are my comments and suggestions:

1. The two supplements (a) Technical appendix (which in some places is apparently called Supplementary materials), and (b) Supplementary results should be labelled and referenced clearly and consistently. Now, for example, it is not always clear in which of the two supplements to search for Tables B.

Response: Apologies that this was not consistent throughout. We have renamed our supplements as Supplementary Methods (technical appendix) and Supplementary Results.

2. The description of the microsimulation model in the online methods requires clarification. In fact, I largely misunderstood the paper after first reading, until I read the very clear description of the methods in the Technical appendix. It may be sufficient to add to the online methods the specification that that it is a discrete time model which simulates individuals' yearly changes in risk factors, health and survival. (The alternative that confused me, when reading the paper first time, was a continuous time model which would simulate the times of change of individuals' status.)

Response: We have added additional detail from the technical appendix to the main methods section of our manuscript:

“We used IMPACT_{NCD}, a validated dynamic discrete-time microsimulation model, to simulate the effect of varying levels of reduction in selected risk factors on future trends in major illness. IMPACT_{NCD} simulates the life course of synthetic individuals aged 30 and over from a close-to-reality English population and translates changes in risk factor exposure trends into changes in annual disease incidence, case fatality, and disease prevalence. Within IMPACT_{NCD}, each unit is a synthetic individual (simulant) represented by a record containing a unique identifier and a set of associated attributes. The microsimulation then projects the life course of each synthetic individual. The attributes of each synthetic individual include sociodemographic characteristics, exposures to risk factors, acquired diseases, and cause of death if relevant. All these attributes are updated in discrete annual steps according to a set of stochastic rules. We structured these rules based on well-established epidemiological principles. Specifically, behavioural risk exposures are conditional on sociodemographic exposures; biological risk exposures are conditional on behavioural and sociodemographic exposures, and diseases are conditional on biological, behavioural, and sociodemographic exposures, as well as diagnosis of other conditions. Finally, mortality depends on sociodemographic, behavioural, biological and disease exposures.” (p18)

3. The base-case scenario of continuing trend has a key role in the results. Therefore, its selection should be justified and discussed. The base-case trend is based on data from much before the projection period, and there is not reason to assume that the risk factor trend would stay linear for forty years. Changes are possible for both directions. Nevertheless, I think the current selection is good because there are no justifiable alternatives. However, if the simulation model allows, I suggest to add a fourth scenario of no trend in risk factors. This would have a clear interpretation and would give an informative comparison in addition to the current scenario of theoretical minimum risk.

Response: Thank you. We assume log-linear trends in risk factor exposure, exposure-response, and disease incidence, and thus, the projected rate of change decreases over time. We have now validated our projections of risk factor trends against more recent health survey for England data (2015-2019), and for all risk factors aside from smoking, our projections are similar to observed risk factor trends and have higher predictive validity than holding the mean constant. For smoking, cigarette smoking rates have decreased faster than our model projects. We are happy to confidently share

this additional validation with the reviewer if we are asked to. Although, of course, this does not guarantee that past trends will continue in the near future, we firmly believe that the continuation of trends has more merit as our baseline scenario versus an alternative in which past trends suddenly stop.

4. Setting single risk factors in turn to the theoretical minimum and assuming that all other risk factors follow the base-case trend is not a very realistic scenario because, in particular, BMI is known to affect SBP. I suggest that, in the main results, the authors consider replacing this assumption with the more plausible assumption which allows SPB to change with BMI. If I understand it correctly, such projections have already been made, and are now reported as “BMI sensitivity” in some of the supplementary tables. This change would make the sum of the contributions of the individual risk factors larger than the contribution of all risk factors together, but this is the reality because of the associations between the risk factors.

Response: Thank you. For the 10% improvement scenario we have moved the ‘BMI sensitivity’ analysis (allowing SBP to change with BMI) to the main analysis, as we agree that this is a more realistic assumption.

We have not changed the theoretical minimum risk scenarios as these are largely illustrative. In addition, the scenario with all risk factors reduces already incorporates the benefits of any additional indirect effects.

5. A major determinant of the results is the changing age structure of the English population. It would be helpful for the reader to add figures of the (observed or projected) age structure in selected years (e.g. 2023, 2033 and 2044). This could go to the Supplementary results.

Response: We have added base-case scenario plots of the age structure for 2023, 2033, and 2043 into the supplementary results, and referred to these plots in the main text.

“The projected population structures for selected years under the base-case scenario are presented in Supplementary Results Section 2.” (p8)

Specific comments (the line numbers refer to the pdf version of the documents provided to the reviewers):

6. Introduction: Where should references 8-10 go, if relevant?

Response: Thank you for spotting this, we have updated our reference list.

7. Lines 75 and 541: Environmental tobacco smoke (ETS) is not mentioned elsewhere in the manuscript, and it is only in the tables of risk factor trends in the Supplementary results. Was its impact really estimated in the analysis? If it was, the results should be reported.

Response: Environmental tobacco smoke (ETS) exposure is calculated for non-smokers based on sociodemographic characteristics and the proportion of smokers within the population. Whilst we do not report the impact of ETS directly, it is included within the

model to capture the indirect effects of smoking prevalence on the population as a whole.

8. Lines 85-86: Figures A.5-A.6 of the Technical appendix do not relate to this. I guess the authors mean Tables B.1-B.2 of the Supplementary results. However, Table B.2 is stratified by age groups, not by birth cohorts.

Response: You are correct, we have updated this to refer to age groups and label the figures correctly.

9. Line 92 and throughout the manuscript: The meaning of the numbers in brackets should be explained? I guess they show the “uncertainty intervals” defined on page 17 of the Technical appendix, which cover some of the uncertainty of the projections.

Response: Thank you. We have added a note the first time we present these: “2 percentage points (pp; 95% uncertainty interval: 1.3pp,2.7pp)”

10. Line 150: Correct “Figure 5” to “Figure 2”.

Response: Thank you, we have updated this.

11. Line 174 and elsewhere: The four quintiles divide the distribution into five parts, usually called the “fifths”. Figure 6 clearly shows results by the fifths rather than on the quintiles.

Response: Thank you. We have amended this to “quintile groups” throughout.

12. Lines 175-180: I suspect over-interpretation of the precision of the projections. The differences are so small that I would rather say “similar” instead of “greater” or “mixed”.

Response: Thank you. Although when presented as differences in percentage point reduction in major illness prevalence these seem like small differences, when translated into absolute numbers of individuals, the differences are meaningful. For example, a scenario of no excess BMI would lead to 153,000 fewer individuals (95% UI: 110,000-223,000) living with major illness in 2043 amongst individuals in the most deprived IMD quintile group, compared to 86,000 (42,000-140,000) fewer individuals living with major illness in 2043 in the least deprived quintile group.

13. Line 221: If sensitivity analysis 1) is the same as described on line 552, the latter description is much clearer.

Response: Thank you. We have amended the former to match the latter: “assuming improved risk factor exposures did not impact case fatality”

14. Lines 244-245: Specifying such details before describing the overall structure of the model is confusing. Adding a reference to the online methods or the technical appendix should be sufficient here.

Response: Thank you. We have added signposting into this sentence: “Our approach (see Methods and Supplementary Methods for details) uses population-attributable risk fractions to translate risk factor exposures to disease incidence and

mortality, and includes lag times between exposure and outcomes.” (p14)

15. Line 258: Substitute “the Health Survey for England (HSE)” for “HSE”.

Response: Corrected

16. Line 312: The missing reference is apparently Supplementary Figure B.1.

17. Line 317: The missing reference is apparently Supplementary Figure B.2.

18. Line 524: The reference is apparently Table 2-1 of the Technical appendix.

19. Line 525: I failed to find model source code in reference 33.

20. Line 528: Shouldn't the reference be 33, not 34,

21. Line 523: Shouldn't the references be “34, 35, 36”.

22. Line 544: I guess the reference should be to the Technical appendix.

Response for 16-22: Thank you – we have updated all our references and cross-references throughout the manuscript

23. Line 549: By “survival” you apparently refer to case-fatality, which is defined in the Technical appendix. Consistent terminology would help the reader.

Response: Thank you. We have replaced “survival” with case-fatality as suggested.

24. Lines 564-565: I find this sentence confusing in the description of methods. This is indicated neither in the title of the paper nor in the objectives in the Introduction.

Response: Thank you. We have amended this for clarity:

“We present the absolute and relative difference in the prevalence rate of major illness within the population (aged 30 and over), compared to a base-case scenario continuing the observed trends in risk factors from 2003-2014.” (P24)

25. Line 566: I suggest to clarify as “... continuing the trends of 2003-2014 in risk factors.”

Response: Thank you. We have amended our description of the base-case scenario to : “compared to a base-case scenario continuing the observed trends in risk factors from 2003-2014” (p24)

26. Line 580: Add a reference to the Technical appendix.

Response: Thank you. We have added a table (Table 3) summarising the scenario implementation to the methods section of the main manuscript.

27. Line 586: I suggest to specify the minimal levels here and add a reference to page 13 of the Technical annex.

Response: Thank you. The aforementioned table summarises these.

Figures: I failed to see some of the light-coloured lines on the figures.

Response: Thank you. We have increased the thickness of the lines, and if this manuscript is accepted for publication will ask the editor for advice.

Reviewer #1 (Remarks on code availability):

As mentioned in my comments to the authors, I failed to find the code in the URL given in the manuscript. I now found the code in your URL. Although I don't have the capacity to review the code, I am happy to see that it is publicly available.

Response: We have added in a code availability statement with the URL.

“Code for model implementation is available here:

https://github.com/ChristK/IMPACTncd_Engl

Reviewer #2 (Remarks to the Author):

The authors present a variety of scenarios in terms of a multiple changes in the prevalence of behavioural and other risk factors on morbidity. The objective is a worthy one but I found the execution in this paper unconvincing.

Response: Thank you for your time in reviewing and commenting on our manuscript.

Structure:

The paper goes from an introduction straight into results. What happened to a methodology section?

Response: Thank you. Our methodology section is after the discussion, following the journal guidelines.

We have amended our final introduction paragraph to give more context for our results:

“The aim of this study is to estimate the effect of improving population risk factor exposures on future trends in major illness and mortality among the overall adult population of England and for population sub-groups living in different areas of deprivation. We used IMPACT_{NCD}, a validated dynamic discrete-time microsimulation model²³⁻²⁷, to simulate the potential impact of two risk factor improvement scenarios on the burden of major illness in England among adults aged 30+ over the next two decades (2023-2043): 1) if the whole population had risk factor exposure levels at the level of estimated theoretical minimum risk; 2) if risk factor exposure levels improved by 10% relative to our base-case scenario. Our base-case scenario continues recent trends in risk factor exposures, as derived from 2003-2014 Health Survey for England (HSE) data. To explore the equity impacts of the scenarios, results for the theoretical minimum risk scenario were stratified by quintile groups of socioeconomic status using the English Index of Multiple Deprivation, an area-level measure of relative deprivation. We focused on eight risk factors (combined into six scenarios): tobacco smoking, environmental tobacco smoke, fruit consumption, vegetable consumption, physical activity, body mass index (BMI), systolic blood pressure (SBP), and total cholesterol. We defined major illness as a score greater than 1.5 in the Cambridge Multimorbidity Score (CMS) – a composite index of 20 long-term conditions weighted based on the impact on the health system and an individual's health outcomes²⁸. IMPACT_{NCD} was informed by data from linked primary care, secondary care, and mortality records, risk factor exposure data from the Health Survey for England, and population estimates and projections from the Office for National Statistics. Further details on the modelling approach can be found in the methods and Supplementary Methods sections.” (P4)

Writing style:

I found the introduction section to be quite tersely written and not an easy read. Even

though the content was fine, it lacked flow.

Response: Thank you. We have revised our introduction and hope that this has improved the flow.

Software:

The results are based on the authors' own IMPACT software tool. This is perhaps fine, but I could not see any reference to the model underpinning the software having been independently peer reviewed in some way.

As a result, when you launch into the Results section with no build up, there is no context for the projected exposure levels. Where did these come from? Where is the historical data to set these up?

And there is no explanation that I could easily see on how the behavioural changes interact with outcomes.

Response: Thank you. Our team developed the open-source IMPACT_{NCD} model, and it has been used in multiple settings over the last decade. Previous versions of the model have been peer-reviewed and published multiple times in high-impact journals, including the BMJ, Circulation, and PLOS Med. We have added references to previous peer-reviews of IMPACT_{NCD} models into the main manuscript, added clearer links to our supplementary methods, and included a brief description of the model at the end of the introduction to provide context for the results:

“The aim of this study is to estimate the effect of improving population risk factor exposures on future trends in major illness and mortality among the overall adult population of England and for population sub-groups living in different areas of deprivation. We used IMPACT_{NCD}, a validated dynamic discrete-time microsimulation model^{23–27}, to simulate the potential impact of two risk factor improvement scenarios on the burden of major illness in England among adults aged 30+ over the next two decades (2023–2043): 1) if the whole population had risk factor exposure levels at the level of estimated theoretical minimum risk; 2) if risk factor exposure levels improved by 10% relative to our base-case scenario. Our base-case scenario continues recent trends in risk factor exposures, as derived from 2003–2014 Health Survey for England (HSE) data. To explore the equity impacts of the scenarios, results for the theoretical minimum risk scenario were stratified by quintile groups of socioeconomic status using the English Index of Multiple Deprivation, an area-level measure of relative deprivation. We focused on eight risk factors (combined into six scenarios): tobacco smoking, environmental tobacco smoke, fruit consumption, vegetable consumption, physical activity, body mass index (BMI), systolic blood pressure (SBP), and total cholesterol. We defined major illness as a score greater than 1.5 in the Cambridge Multimorbidity Score (CMS) – a composite index of 20 long-term conditions weighted based on the impact on the health system and an individual’s health outcomes²⁸. IMPACT_{NCD} was informed by data from linked primary care, secondary care, and mortality records, risk factor exposure data from the Health Survey for England, and population estimates and projections from the Office for National Statistics. Further details on the modelling approach can be found in the methods and Supplementary Methods sections.” (p24(

We have also included additional information in the methods overview section to clarify the relationships between different attributes:

“Within IMPACT_{NCD}, each unit is a synthetic individual (simulant) represented by a record containing a unique identifier and a set of associated attributes. The microsimulation then projects the life course of each synthetic individual. The attributes of each synthetic individual include sociodemographic characteristics, exposures to risk factors, acquired diseases, and cause of death if relevant. All these attributes are updated in discrete annual steps according to a set of stochastic rules. We structured these rules based on well-established epidemiological principles. Specifically, behavioural risk exposures are conditional on sociodemographic exposures; biological risk exposures are conditional on behavioural and sociodemographic exposures, and diseases are conditional on biological, behavioural, and sociodemographic exposures, as well as diagnosis of other conditions. Finally, mortality depends on sociodemographic, behavioural, biological and disease exposures.” (p18)

BMI:

Is mean BMI the sole variable rather than a distribution across all levels of BMI that shifts over time. If it is just the mean then there is a problem with the underlying model. The reason for this is that the risk of developing disease has a non-linear relationship with BMI. So you really need to model separately e.g. the % who are 1: morbidly obese or 2: obese or 3: overweight etc.

The same might go for other risk factors.

Note that there is difference between BMI which is effectively an intermediate risk factor whereas other risk factors such as smoking, fruit and veg, and activity are behavioural risk factors more in the control of the individual.

Response: Thank you. We are using shifting distribution for all the exposures in the model. We describe our approach in detail in the Technical Appendix, pp8-9 and pp132-135.

Other:

On page 6 and several other places there is a very obvious (i.e. in bold) "Error!". This shows that the authors have not checked the version submitted and is somewhat disrespectful towards the journal and referees.

Response: We apologise for this oversight. These errors are corrected in this updated version.

Reviewer #3 (Remarks to the Author):

This paper utilizes the validated microsimulation model (IMPACTNCD) to project the potential effects of improving population-level exposure to eight risk factors on the burden of major illness among adults aged 30 and older in England from 2023 to 2043. While the study offers valuable insights into how risk factor improvements could shape future health outcomes, several limitations and areas requiring clarification remain, which could enhance the methodological rigor and interpretability of the findings:

Response: Thank you for your review, we hope that our revisions address your requests for clarification.

1. The selection of eight risk factors is not fully justified. Important contributors to chronic diseases, such as alcohol consumption and air pollution, are excluded, raising concerns about the completeness of the projections. The authors should explain the rationale for focusing on these specific risk factors. Whether the choice was influenced by data limitations in the Health Survey for England (HSE)?

Response: Thank you. The methodology for measuring current alcohol exposure in HSE has changed over the years, making extracting trends unreliable. Moreover, past exposure to alcohol was not measured at all, creating distortions in our modelling approach. Therefore, we decided not to present results for alcohol.

Air pollution may be substantial in hotspot areas and for some subpopulations, but overall, its impact is small in comparison to the other risk factors modelled. Furthermore, concentrations of pollutants in the atmosphere that are regularly monitored are only indicative, at best, of personal exposure to air pollutants, as personal exposure is affected by a multitude of factors. We are unaware of any nationally representative dataset that includes personal exposure measurements to air pollution. Therefore, at the moment, we do not include air pollution in our model.

We summarise the above and comment on the choice of risk factors in the discussion and methods sections:

Discussion:

“We modelled the direct effect of risk factors on disease incidence where high-quality, robust evidence was available. We did not model causal associations of other factors (e.g. sugar intake or alcohol consumption) for which measurement was unavailable or was inconsistent in the Health Survey for England (HSE); there are, therefore, possible additional benefits to be gained by tackling these risk factors. For example, we did not model exposure to air pollution. However, the impact of air pollution is small compared to the other risk factors modelled³⁰.” (p14-15)

Methods:

“The direct causal relationships between amenable risk factors and disease incidence included in the model were chosen based on the availability of high-quality evidence from systematic reviews and meta-analyses and are displayed in Table 2.” (P21)

2. The manuscript does not explicitly address whether interactions or mediating effects between risk factors (e.g., BMI influencing total cholesterol or systolic blood pressure) were considered in the model. Given the complexity of chronic disease etiology, such interactions are critical. The authors should clarify whether these effects were incorporated, describe how they were modeled if applicable, and discuss the implications for the projections if they were not.

Response: Thank you. We have modelled the indirect effect of decreasing BMI through lowering SBP and total cholesterol, which we now present in our main analysis.

“For the 10% improvement in BMI scenario, we modelled the indirect effect of decreasing BMI on SBP and total cholesterol: for BMI \geq 25, for each 1 unit decrease in BMI, we modelled a 0.483mmol/L (95% confidence interval (CI): 0.330, 0.649) decrease

in total cholesterol⁵³, and a 2.29mmHg (95% CI: 1.56, 3.55) decrease in SBP⁵⁴. As a sensitivity analysis, we also modelled a direct effect-only BMI scenario without these indirect effects on SBP and total cholesterol.” (p25)

3. It is unclear whether the 26 diseases modeled correspond to all the long-term conditions listed in the Cambridge Multimorbidity Score (CMS). The authors should clarify this alignment and provide specific data or a comparison to substantiate their selection. Additionally, the advantages of focusing on these 26 diseases should be discussed.

Response: Thank you. Yes, they do. We have added a table (table 2) into the methods showing how the 26 modelled conditions relate to the 20 conditions in the Cambridge Multimorbidity Score (CMS). This also demonstrates the rationale for modelling some conditions separately e.g. the key risk factors for breast cancer and prostate cancer are different.

We chose the CMS index as these conditions comprise 65% of disability-adjusted life years in England, and the condition weights are based on how the illness is likely to affect patients’ use of primary care, emergency health services and likelihood of death.

4. Although the manuscript mentions that the IMPACTNCD model is well validated, the main text lacks details on the validation process and results. This reliance on supplementary materials makes it difficult for readers to access critical information. Including a concise summary of key validation outcomes, such as predictive accuracy, error margins, or comparisons with other models, in the main text would improve the transparency and credibility of the study.

Response: Thank you. We have added a sub-section on validation into the methods of the main manuscript.

“Model validation and calibration

We validated the IMPACT_{NCD} epidemiological engine using internal validation, plotting the modelled exposures’ prevalence and disease incidence against the observed exposures’ prevalence and disease incidence in HSE and linked primary care data, respectively. Mortality in the model is calibrated to ONS mortality projections (see Supplementary Methods – Mortality Calibration section (p16)). Our risk factor projections follow similar patterns to observed values from HSE stratified by year and age group and by quintile groups of IMD and age group (Supplementary Methods p137-175). For our primary outcome prevalence of CMS > 1.5 (defined as ‘major illness’ in this paper), we plotted projected major illness against the observed CPRD Aurum prevalence (see Supplementary Methods Figure 5-1 (p19)). Validation plots for incidence, case fatality, and prevalence of individual conditions are shown in the Supplementary Methods (p40-132). Overall, the validation plots suggest that IMPACT_{NCD} captures exposure trends and translates them to disease incidence and mortality reasonably well for the purpose of this project.” (p26)

NCOMMS-24-70452A Response to Reviewer comments

Reviewer #1 (Remarks to the Author):

The authors have addressed satisfactorily most of my comments. I only want to comment on the responses to three of my earlier comments:

3. I agree with the response, but it does not address my earlier suggestion. (Perhaps my comment was unclear). I was not questioning the use of continuing trends as the baseline scenario. I suggested to add the interesting scenario of no trend. I think is even more realistic than the scenario of theoretical minimum risk, which is already included in the paper. I hope the authors consider this suggestion, but I don't insist on it.

Response: Thank you for taking the time to re-review our paper. We indeed misunderstood your earlier suggestion. Adding another set of scenarios to this paper might render it too dense and complex for a wider readership to digest. That said, we concur that these scenarios would be intriguing, and we will aim to simulate them in the future, publishing the results on an institutional webpage or in an academic publication.

According to the response, the authors have now validated the projections of risk factor trends against a more recent health survey for England data (2015-2019). Why not mention this valuable information on the paper?

Response: Thank you. We are preparing to publish this validation as part of a separate manuscript. We would be pleased to submit this to Nature Communications for initial consideration. In the meantime, if editorial policy permits, we are content to add one sentence to our discussion summarising this validation exercise without presenting the results.

12. The authors' response highlights the importance of my concern but does not remove the problem: The reported differences in percentage points correspond to large numbers of persons with major illness, but comparison of the provided uncertainty intervals does not convince the reader about real differences. The authors should recognize this fact in the paper.

It is possible that the comparison of the IMD groups is actually more precise than suggested by the reported uncertainty intervals. If this is the case, it should be stated in the paper.

Response: Thank you. The scenarios are not statistically independent, by design, so 1) it is not straightforward to mentally extrapolate the uncertainty interval of a comparison between scenarios from the uncertainty of the comparands, and 2) overlapping uncertainty intervals do not indicate a 'lack of statistical significance'. The concept is similar to that of paired study designs, where overlapping confidence intervals do not necessarily indicate a lack of statistical significance. In this revised version of our

manuscript, we have calculated the appropriate uncertainty intervals for the difference between the deprivation groups to make this section clearer. It now reads:

“For BMI, greater absolute reductions are seen among people living in the most deprived compared to the least deprived quintile group. For example, a scenario of no excess BMI would reduce major illness prevalence amongst individuals in the most deprived IMD quintile group by 0.4pp (0.1pp,0.6pp) more compared to the least deprived fifth. This socioeconomic gradient was observed in 99% of our simulations. Conversely, theoretical minimum risk levels of SBP result in greater decreases in major illness prevalence among people living in the least deprived quintile group (by 0.4pp (0.1pp,0.6pp) more compared to the most deprived IMD quintile group. The socioeconomic gradients were small for physical activity, smoking, and total cholesterol; however, for all four of these factors, the reduction in major illness prevalence was more likely to be greater in the least deprived quantile. We did not observe a gradient for the fruit and vegetable scenario. In particular for smoking, its elimination could lead to an increase in major illness prevalence among those living in the most deprived IMD quintile group, resulting from greater decreases in all-cause mortality than in major illness incidence - people living longer but at risk of more years in ill health.”

23. Third paragraph of the Discussion: Shouldn't some to the words "mortality" be replaced with "case fatality"?

Response: Thank you for spotting this, we have amended it to case fatality.

“Our approach (see Methods and Supplementary Methods for details) uses population-attributable risk fractions to translate risk factor exposures to disease incidence and case fatality and includes lag times between exposure and outcomes.”

Reviewer #3 (Remarks to the Author):

Thank you for your responses. Regarding point 2, I appreciate the clarification on BMI's indirect effects. However, my concern extends beyond this specific example. Chronic disease risk factors often interact in complex ways, which could influence disease progression and overall projections. Could the authors clarify whether additional interactions or mediating effects were considered in the model? If these were not explicitly accounted for, a brief discussion of this limitation and its potential impact on the findings would be valuable.

Response: Thank you for re-reviewing our paper. We agree that chronic disease risk factors may interact in complex ways. However, epidemiological studies are rarely designed to capture this complexity in a robust and quantifiable manner that would allow us to simulate it robustly. For example, the structure of the regressions typically used in epidemiological studies often masks such complexities. We argue that we went beyond what other models have considered in similar modelling exercises by 1) modelling the clustering of risk factors to segments of the population, 2) modelling the

lag between exposure and disease, 3) allowing for the cumulative effect of smoking on some diseases, 4) allowing risk factors to affect incidence and case fatality and 5) allowing diseases to be risk factors of other diseases. Nevertheless, we applied some assumptions to our model that simplify the potential complexity. For instance, we assumed multiplicity of risks when multiple risk factors were present. We discuss all these points in various sections of the technical appendix, and we summarise the key modelling assumptions in Table 1 of the main text, along with references to the technical appendix for further explanation.

We have added at the beginning of our limitations paragraph the underlined sentence

“Our study has some limitations. We modelled the direct effect of risk factors on disease incidence where high-quality, robust evidence was available. However, epidemiological studies may oversimplify the complex interplay between risk factors and diseases.”

We additionally added to our discussion section the phrase.

“In other words, the mixture and severity of conditions among people with major illness might change as a result of the improvements in risk factor exposures, but CMS and our definition of major illness are insensitive to these changes.”

Reviewer #4 (Remarks to the Author):

I thank the authors for their concerted efforts in addressing the comments provided by all three reviewers. In particular I note that the Introduction is substantially improved, and now provides a readable yet compact examination of this research area and methodology.

I also commend the authors' work on the Discussion, which delineates very clearly the limitations of this study and the data used to inform the model, and concludes with a cogent analysis of the difficulties of developing and implementing evidence-based policy for the reduction of major illness. My two immediate concerns with this study -- that the data originated from before the Covid-19 pandemic, and that behavioural changes in response to diagnoses were not modelled -- were appropriately called out in this section.

I feel this study also makes an implicit case for the benefits of applying multiple simulation approaches to major population health challenges. Some of the details lacking here (e.g., the behavioural changes in response to diagnosis, unmet need, etc.), can be captured in agent-based models using appropriate behavioural assumptions vetted by domain experts (which is what I do, incidentally). Conversely, the microsimulation work enhances related complex-systems-based methods like ABM by providing projections and outcomes that generate policy ideas, which can then be tested and examined in the ABM framework. In future work, I feel it would be productive to seek out collaborative relationships with research groups using such methods to cover the gaps in microsimulation, and in turn, to assist the ABM researchers in

constructing their models with greater realism and precision, which is often very difficult in ABM work.

Response: Thank you for your time in thoughtfully reviewing our paper. We agree that it would be productive for ABM and microsimulation modellers to work together in collaboration to strengthen the application of these methods.

The current sensitivity analysis does not provide detailed uncertainty quantification nor does it quantify the level of contribution of different model parameters to the final output variance. More sophisticated techniques like Gaussian process emulation or similar provide these outputs, and are useful for model calibration as well as sensitivity analysis and uncertainty quantification.

Response: Thank you for these insightful comments. The limiting step in these techniques is the requirement for substantial computational resources to either provide the necessary inputs or implement them effectively with the current version of the model. We are working on a more efficient version of IMPACTncd, which, once completed, may enable us to produce, among other things, tornado plots and value of information analysis. However, the 2nd-order Monte Carlo probabilistic sensitivity analysis we currently implement is the current ceiling.

Overall, I feel the manuscript is providing a useful examination of major health challenges in England, which are illuminating for policymakers and other modellers, and the provision of the code under the GPL licence is very helpful for the modelling community in this area. I believe the authors have adequately modified and clarified the paper to address the concerns of all three reviewers.

I would make only two very small additional requests: firstly, line 142 is missing the 'P' in 'Projections'; and Section 3 of the GitHub repository's documentation should be completed with appropriate references to prior publications using ImpactNCD for the benefit of future users prior to publication of this paper.

Response: Thank you for spotting these. We have amended line 142 and updated the GitHub repository documentation.

Reviewer #4 (Remarks on code availability):

The code provided in the GitHub repository is provided with detailed instructions for installation including screenshots, so the model is very easy to install. The code appears to run as advertised; however, as I do not use R for simulation modelling but instead general-purpose programming languages like Python and Julia, I cannot evaluate the readability of the code itself.

I do note however that Section 3 of the ReadMe on the repository (Further Notes and References) is not filled out. This should be done so that potential users can refer back to previous published works produced using this modelling framework.

Response: We have updated the ReadMe of our GitHub repository to include further notes and references.

NCOMMS-24-70452C Response to Reviewer comments

REVIEWERS' COMMENTS

Reviewer #1 (Remarks to the Author):

I am happy with the Authors' response to my earlier comments, except a small terminology issue in the Results. The subsection "Impact on inequalities" uses terms "quintile group", "fifth" and "quantile" for the same issue. To make the paper easier to read, I suggest the authors unify the terminology (throughout the paper). I prefer the term "fifth", but would also be happy with "quintile group" which is a new term to me. "Quantile" (or more specifically "quintile") refers to a point and therefore is incorrect.

Response: Thank you for further reviewing our manuscript. We now use 'quintile group' throughout.

Reviewer #3 (Remarks to the Author):

The authors have improved the manuscript.

Response: Thank you for taking the time to re-review our manuscript.

Reviewer #4 (Remarks to the Author):

Having read through the authors' rebuttal letter and perusing the updated manuscript and code repository, I'm pleased to say that the authors seem to have taken all my comments into account. I can see that significant text has been added/changed to reflect the comments of the other reviewers as well, which has improved the study by making the limitations of the methods more clear, as well as clarifying some of the terminology used.

From my perspective, the work is a scientifically sound piece of microsimulation research which provides useful insight into the contribution of various risk factors to major illness in England. I feel the study will provide value to policymakers as well as researchers, who will be able to access and built upon the open-source code provided. Given that my comments and concerns were addressed, I would therefore recommend that this version of the manuscript be approved for publication.

Dr Eric Silverman
University of Glasgow

Reviewer #4 (Remarks on code availability):

The code is well-documented and runs well. As someone who uses general-purpose programming languages like Python, Julia and C rather than R, I cannot comment on the readability of the code itself. In my previous review, I noted that the readme on the GitHub repository was missing some valuable context and links to previous works; this has now been rectified.

From my perspective the authors have done a fine job in documenting their code and ensuring that potential users can easily make use of it. Having said that, I personally prefer using more open licensing than the GPL (Creative Commons or the MIT License), given that our work is publicly funded, but the GPL is good enough.

Response: Thank you for such a detailed review of our work, your suggestions helped us improve our manuscript.